# Swimming motility of a gut bacterial symbiont promotes resistance to intestinal expulsion and enhances inflammation

**Travis J. Wiles**[1][○], **Brandon H. Schlomann**[1,2][○], **Elena S. Wall**[1], **Reina Betancourt**[1],
**Raghuveer Parthasarathy**[1,2], **Karen Guillemin**[1,3]*

**1** Institute of Molecular Biology, University of Oregon, Eugene, Oregon, United States of America,
**2** Department of Physics and Materials Science Institute, University of Oregon, Eugene, Oregon, United
States of America, **3** Humans and the Microbiome Program, CIFAR, Toronto, Ontario, Canada

○ These authors contributed equally to this work.
* kguillem@uoregon.edu

pbio.3000661

for Biological Studies, UNITED STATES

**Data Availability Statement:** All relevant data are
provided within the paper and its Supporting
Information files. All data values plotted in figures

## Abstract

Some of the densest microbial ecosystems in nature thrive within the intestines of humans
and other animals. To protect mucosal tissues and maintain immune tolerance, animal
hosts actively sequester bacteria within the intestinal lumen. In response, numerous bacte-
rial pathogens and pathobionts have evolved strategies to subvert spatial restrictions,
thereby undermining immune homeostasis. However, in many cases, it is unclear how
escaping host spatial control benefits gut bacteria and how changes in intestinal biogeogra-
phy are connected to inflammation. A better understanding of these processes could
uncover new targets for treating microbiome-mediated inflammatory diseases. To this end,
we investigated the spatial organization and dynamics of bacterial populations within the
intestine using larval zebrafish and live imaging. We discovered that a proinflammatory *Vib-
rio* symbiont native to zebrafish governs its own spatial organization using swimming motility
and chemotaxis. Surprisingly, we found that *Vibrio*'s motile behavior does not enhance its
growth rate but rather promotes its persistence by enabling it to counter intestinal flow. In
contrast, *Vibrio* mutants lacking motility traits surrender to host spatial control, becoming
aggregated and entrapped within the lumen. Consequently, nonmotile and nonchemotactic
mutants are susceptible to intestinal expulsion and experience large fluctuations in absolute
abundance. Further, we found that motile *Vibrio* cells induce expression of the proinflamma-
tory cytokine tumor necrosis factor alpha (TNFα) in gut-associated macrophages and the
liver. Using inducible genetic switches, we demonstrate that swimming motility can be
manipulated in situ to modulate the spatial organization, persistence, and inflammatory
activity of gut bacterial populations. Together, our findings suggest that host spatial control
over resident microbiota plays a broader role in regulating the abundance and persistence
of gut bacteria than simply protecting mucosal tissues. Moreover, we show that intestinal
flow and bacterial motility are potential targets for therapeutically managing bacterial spatial
organization and inflammatory activity within the gut.

(main and supplemental) are tabulated in the supplemental spreadsheet S1 Data.

**Funding:** Research was supported by an award from the Kavli Microbiome Ideas Challenge, a project led by the American Society for Microbiology in partnership with the American Chemical Society and the American Physical Society and supported by The Kavli Foundation. Work was also supported by the National Science Foundation under Awards 1427957 (R.P.) and 0922951 (R.P.), the M.J. Murdock Charitable Trust, and the National Institutes of Health (http://www. nih.gov/), under Awards P50GM09891 and P01GM125576 to K.G. and R.P., F32AI112094 to T.J.W., and T32GM007759 to B.H.S. R.B. was supported by National Science Foundation BIO/DBI Award 1460735 as a visiting undergraduate intern. The University of Oregon Zebrafish Facility is supported by a grant from the National Institute of Child Health and Human Development (P01HD22486). The funders had no role in study design, data collection and analysis, decision to publish, or preparation of the manuscript.

**Competing interests:** The authors have declared that no competing interests exist.

**Abbreviations:** Aer, *Aeromonas*; aTc, anhydrotetracycline; bp, base pair; cvz, conventionalized; CRISPRi, CRISPR interference; dCas9, dead Cas9; EM, embryo media; ESEM, environmental scanning electron microscopy; gf, germ-free; GFP, green fluorescent protein; GOF, gain-of-function; hpi, hours post inoculation; LOF, loss-of-function; LSFM, light sheet fluorescence microscopy; ns, not significant; OD, optical density; PCR, polymerase chain reaction; RBS, ribosome binding site; rs, restriction site; sfGFP, superfolder green fluorescent protein; sgRNA, single-guide RNA; sib, sibling controls; SOE, splice by overlap extension; TetR, Tet repressor protein; TNFα, tumor necrosis factor alpha; UTR, untranslated region; wt, wild type.

# Introduction

Humans and other animals foster diverse microbial communities within their intestines. Although these symbiotic consortia support vital aspects of host biology, they can also harbor proinflammatory pathogens and pathobionts, which are indigenous members of the microbiota that have latent pathogenic potential [1,2]. Understanding how hosts normally constrain the virulent activities of resident bacteria and the mechanisms by which disease-causing lineages escape this control will open new opportunities for developing microbiome-based therapies to improve human and animal health.

One way hosts keep potentially pathogenic bacterial lineages in check within the intestine is by imposing widespread restrictions on microbiota spatial organization. The most recognized spatial control measures employed by the host are mucus, immunoglobulins, and antimicrobial peptides, which act to confine bacteria to the intestinal lumen, away from mucosal surfaces [3–5]. In turn, it is thought that intense competition for resources pushes bacteria to evolve strategies for subverting host control and occupying new spatial niches [6,7]. In line with this idea, several prototypic pathobionts undergo blooms in abundance that are coincident with shifts in intestinal biogeography [1,8–10]. A potential trait underlying this behavior that is common to many pathobionts—as well as numerous bona fide pathogens—is flagella-based swimming motility [11–13].

Swimming motility, together with chemotaxis, gives bacteria the agency to govern their own spatial organization and access niches that are typically thought to enhance growth and survival [14–17]. The connection between motility and gastrointestinal colonization has historically been studied in the context of pathogens such as *Helicobacter pylori*, *Campylobacter jejuni*, *Vibrio cholerae*, and *Salmonella* Typhimurium [12]. With *S.* Typhimurium in particular, it has been found that this pathogen uses motility and chemotaxis to associate with and invade the intestinal mucosa, induce inflammation, and facilitate growth [17–19]. In several instances, it has also been shown that flagellin, the protein subunit comprising the bacterial flagellum, can be a major driver of inflammation [20–22]. Highlighting the importance of curbing the pathogenic potential of motile bacteria, studies in mice have revealed several mechanisms by which hosts detect and quench flagellar motility to maintain intestinal homeostasis [4,20,23,24]. In total, significant progress has been made in understanding the role motility plays in the infectious lifestyles of pathogens and its proinflammatory consequences for the host. However, a broader view of how motility behaviors might be shaping the lifestyles of resident gut bacteria remains limited and largely unexplored.

Insights into this question have started to emerge from our studies on how diverse bacterial taxa colonize the zebrafish intestine. The optical transparency and small size of larval zebrafish make them an ideal vertebrate model for probing how bacteria use motility to spatially organize their populations within a living animal. With light sheet fluorescence microscopy (LSFM) it is possible to capture the full three-dimensional architecture of bacterial populations at single bacterial cell resolution across the entire length of the larval intestine [25]. In addition, the spatiotemporal dynamics of bacterial and host cells can be followed in real time or over the course of many hours. Using LSFM, we have found that for many noninflammatory commensal bacteria native to the zebrafish microbiome, the bulk of their populations are nonmotile and reside as dense multicellular aggregates within the intestinal lumen [26,27]. Notably, this pattern of bacterial spatial organization is consistent with histological data from both the mouse and human intestine [28–31]. We discovered that in this aggregated regime, bacteria are extremely vulnerable to intestinal flow. Consequently, aggregated bacterial populations can be stochastically expelled from the host in large numbers, producing punctuated drops in abundance [32,33].

In contrast, unlike most zebrafish gut bacteria studied thus far, we have identified an isolate of nontoxigenic *V. cholerae* (strain ZWU0020, further referred to as "*Vibrio*" for brevity) that exhibits pathobiont-like characteristics and assembles intestinal populations made up of planktonic cells displaying vigorous swimming motility [32,34]. This particular *Vibrio* strain was originally isolated from the intestine of an apparently disease-free zebrafish and does not encode cholera toxin or toxin-coregulated pilus [35]. The mass swimming behavior of *Vibrio* populations gives them a liquid-like space-filling property that promotes frequent and close contact with the intestinal mucosa [32]. This attribute appears to make *Vibrio* highly resistant to intestinal expulsion. As a result, *Vibrio* stably colonizes the intestine and reaches absolute abundances that are up to 10 times higher than other zebrafish symbionts [33]. *Vibrio*'s unique intestinal lifestyle is also potentially linked to its pathobiont character, which is marked by its ability to supplant established, naturally aggregated bacterial populations [32], and induce intestinal inflammation and exacerbate pathology in susceptible hosts [34,36].

In the present work, we used *Vibrio* as model gut symbiont to investigate the mechanisms by which its motility behaviors control its colonization and contribute to its proinflammatory potential. By combining live imaging, host and bacterial mutants, and in situ manipulation of motility behaviors, we were able to disentangle *Vibrio*'s requirements for motility during multiple stages of intestinal colonization. We found that, for *Vibrio*, swimming motility and chemotaxis do not enhance exponential growth rate but rather enable cells to physically resist intestinal expulsion and maintain stable, highly abundant populations. We also found that host tissues—namely, gut-associated macrophages and cells within the liver—are acutely sensitive to bacterial motility and spatial organization within the intestine. Together, our work expands the scope of bacterial swimming motility during intestinal colonization by revealing how motility can shape the large-scale spatial organization and dynamics of gut bacterial populations. Our study further shows that intestinal mechanics are a host spatial control measure capable of regulating the abundance and persistence of gut bacteria. Ultimately, our work yields new mechanistic insights into the form and function of the intestinal ecosystem and highlights that bacterial motility and the factors controlling the spatial organization of resident microbiota are potential targets for therapeutic manipulation of the gut microbiome.

## Results

### Loss of swimming motility or chemotaxis attenuates intestinal colonization and interbacterial competition

To dissect the role of flagellar motility during intestinal colonization, we generated 2 motility-deficient *Vibrio* mutants (S1A Fig). To test swimming motility in general, we deleted the 2-gene operon *pomAB* that encodes the polar flagellar motor (creating a swimming motility-deficient *Vibrio* mutant we refer to as "Δmot"). To test *Vibrio*'s ability to spatially organize its populations in response to environmental cues, we deleted the gene *cheA2*, which encodes a histidine kinase necessary for chemotaxis (creating a chemotaxis-deficient *Vibrio* mutant we refer to as "Δche"). In vitro, Δmot exhibited complete loss of swimming motility, whereas Δche had a run-biased behavior with swim speeds comparable to wild type but failed to chemotax in soft agar (S1B Fig). Both motility mutants displayed normal growth and assembled a single polar flagellum similar to wild type (S1C and S1D Fig).

We first assessed the absolute abundances of each strain over time by gut dissection and cultivation. We inoculated equal amounts of wild-type *Vibrio*, Δmot, and Δche individually into the aqueous environment of 4-day-old germ-free larval zebrafish. *Vibrio* rapidly colonized germ-free animals to high abundance, reaching a maximal carrying capacity of $10^5$ to $10^6$ cells per intestine by 24 hours post inoculation (hpi) and maintaining a high level of abundance

through 72 hpi (Fig 1A). In contrast, Δmot and Δche displayed attenuated intestinal colonization phenotypes (Fig 1A). Both mutants were slow to access the zebrafish intestine and reached maximal abundances at 24 hpi that were 10- to 100-fold lower than wild type (Fig 1A). This observation suggests that each mutant has a reduced immigration rate, which is in line with previous work indicating that bacterial motility traits may facilitate dispersal and initial colonization of the zebrafish intestine [37,38]. Importantly, differences in intestinal abundances did not appear to be due to differences in the water environment because the levels of each strain outside the fish remained constant and comparable over the assay period (Fig 1A inset).

We next compared the ability of wild-type *Vibrio* and each mutant to invade an established population of *Aeromonas veronii* (strain ZOR0001, further referred to as "*Aeromonas*"). Like *Vibrio*, *Aeromonas* species are abundant members of the zebrafish intestinal microbiota [35]. Previous studies suggest that these 2 genera naturally compete against one another within complex intestinal communities [39]. In addition, we have shown that *Vibrio* is capable of invading and displacing established *Aeromonas* populations in gnotobiotic animals [32]. Following the competition scheme depicted in Fig 1B, we found that each *Vibrio* strain had a distinct competitive interaction with *Aeromonas* (Fig 1C). Wild-type *Vibrio* potently colonized *Aeromonas*-occupied intestines and induced 10- to 100-fold drops in *Aeromonas* abundances (Fig 1C). Zebrafish colonized with the Δmot mutant, however, were dominated by *Aeromonas*, which did not experience any significant declines in abundance compared to monoassociation (Fig 1C). Invasion with the Δche mutant had an intermediate impact on *Aeromonas* abundances and the two appeared to coexist (Fig 1C). Comparing abundances during competition to those during monoassociation showed that each *Vibrio* strain's colonization was hindered to varying degrees while invading established *Aeromonas* populations (Fig 1D). Wild-type *Vibrio* abundances were only 2-fold lower during competition than during monoassociation (Fig 1D). In contrast, Δmot abundances were 6-fold lower (and in several instances reduced by up to 100-fold), whereas the impact on Δche abundances was intermediate with a 4-fold reduction (Fig 1D). Overall, these data show that *Vibrio* requires swimming motility and chemotaxis for normal intestinal colonization and interbacterial competition.

## Motility and chemotaxis mutants have altered intestinal spatial organization

We previously showed that wild-type *Vibrio* cells strongly localize to the larval zebrafish foregut (Fig 2A and 2B) [27], which is an anatomical region comparable to the mammalian small intestine (namely, the duodenum and jejunum) [40,41], and display a highly active swimming behavior both within the intestinal lumen and at mucosal surfaces [32]. In contrast, zebrafish symbionts naturally lacking motility within the gut, like *Aeromonas*, form populations that display a posterior-shifted spatial distribution and a high degree of lumenal aggregation [27,32]. Thus, the impaired competitiveness of Δmot and Δche against *Aeromonas* (Fig 1C) could be associated with changes in their intestinal spatial organization.

To determine how motility and chemotaxis affect *Vibrio*'s cellular behavior and spatial organization within the intestine, we examined wild type, Δmot, and Δche in live animals using LSFM. A fluorescently marked variant of each strain was first monoassociated with germ-free animals and then imaged at 48 hpi. In line with our previous characterizations [26,27], wild-type *Vibrio* assembled dense populations concentrated within the foregut that were almost entirely composed of planktonic cells swimming in the lumen as well as within the intestinal folds (Fig 2C–2G, S1 and S2 Mov). A movie representation of the static image presented in Fig 2C "wt" under "foregut region" was previously published and can be viewed here: https://doi.org/10.6084/m9.figshare.7040309.v1 [26]. In contrast, Δmot and Δche assembled populations with greatly altered behavior and spatial organization.

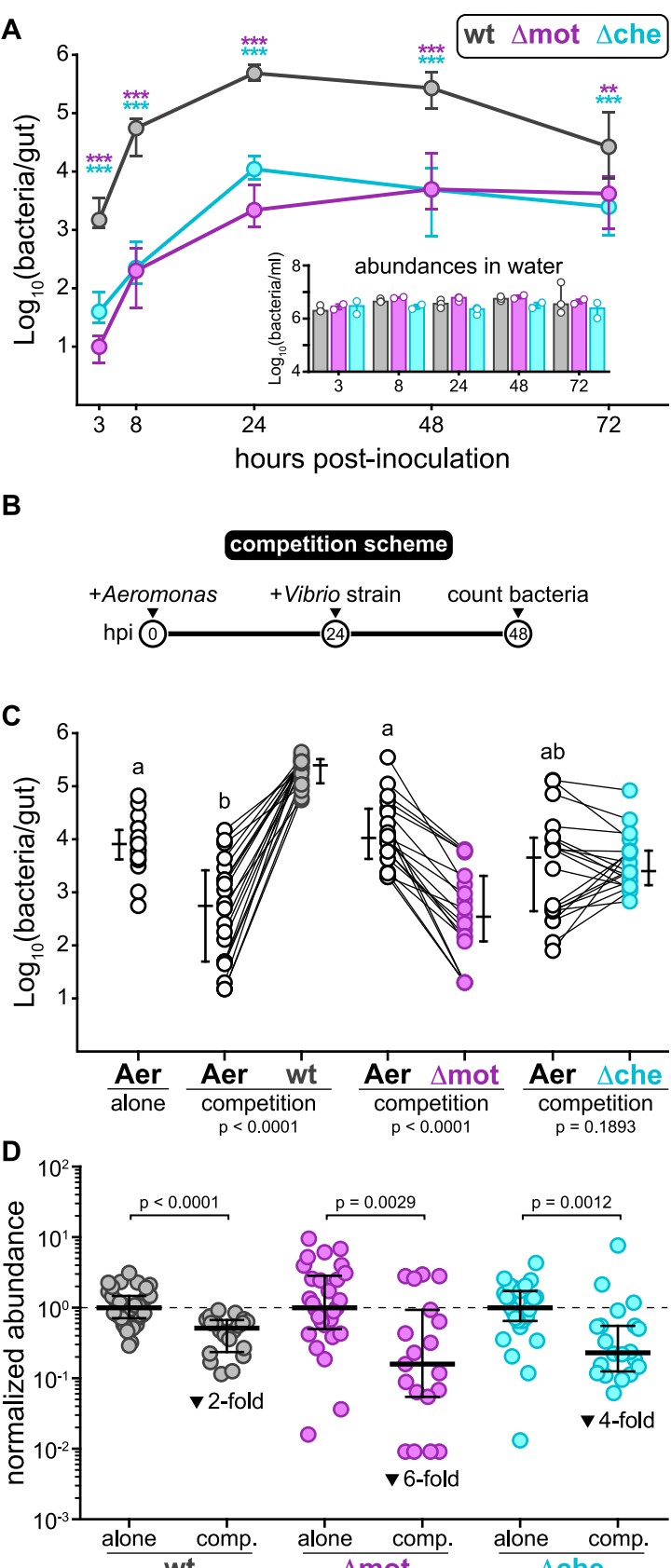

**Fig 1. Loss of swimming motility or chemotaxis attenuates intestinal colonization and interbacterial competition.**
(A) Abundances of wt *Vibrio*, Δmot, and Δche during monoassociation. Plotted are medians and interquartile ranges
($n \geq 17$ animals/marker). Significant differences between each mutant and wt *Vibrio* determined by Mann-Whitney
(purple asterisks: Δmot; cyan asterisks: Δche). ***$p < 0.0001$, **$p = 0.0002$. Inset shows median bacterial abundances in
the water environment from each replicate experiment across all time points. (B) Experimental timeline of
*Aeromonas*–*Vibrio* competition. (C) Intestinal abundances of *Aeromonas* and wt or mutant *Vibrio* strains during
different competition schemes. *Aeromonas* abundances while alone during monoassociation are shown for statistical
comparison. Letters denote significant differences between *Aeromonas* treatments. Lines show paired *Aeromonas* and
*Vibrio* abundances within individual fish. $p < 0.05$, Kruskal-Wallis and Dunn's multiple comparisons test. Adjacent
bars denote medians and interquartile ranges. Significant differences based on Wilcoxon between *Aeromonas* and each
*Vibrio* strain are noted below each competition. (D) Abundances of wt and mutant *Vibrio*s during competition with
*Aeromonas* (from panel C) normalized to abundances during monoassociation at 24 hpi (from panel A). Bars denote
medians and interquartile ranges. Significant differences determined by Mann-Whitney. Fold-decreases based on
medians. Underlying data plotted in panels A, C, and D are provided in S1 Data. Aer, *Aeromonas*; hpi, hours post
inoculation; wt, wild type.

Populations of Δmot were nonmotile whereas Δche had a small subset of motile cells that
could often be observed in the foregut (Fig 2C, S1 Mov). Both Δmot and Δche also became
highly aggregated within the intestine (Fig 2C–2E) despite exhibiting no signs of aggregation
during in vitro culture (S1B and S1C Fig). In S1 Mov we provide a live representation of Δche
within the midgut to emphasize its aggregated form within this intestinal region. The fraction
of planktonic cells contained within each mutant population within the intestine was >10-fold
lower than wild type (Fig 2F). The aggregated cells of Δmot appeared to be mostly restricted to
the lumen, whereas the swimming cells of Δche, like wild type, were observed within the intes-
tinal folds (Fig 2D and 2E, S2 Mov). The Δmot mutant was also largely excluded from the ante-
rior most portion of the foregut, whereas Δche often formed a layer of cells associated with the
anterior wall near the esophageal-intestinal junction (Fig 2C). The population-wide aggrega-
tion of both mutants (which we refer to as cohesion) coincided with an overall posterior-shift
in distribution compared to wild type (Fig 2C and 2G). This shift in distribution is consistent
with previous findings of strong correlations across bacterial species between cohesion and
localization along the zebrafish intestine [27]. In total, our live imaging data show that *Vibrio*
requires swimming motility and chemotaxis to spatially organize its populations within the
intestine. Further, Δmot and Δche formed aggregated and lumen-restricted populations remi-
niscent of zebrafish bacterial symbionts that largely lack swimming motility in vivo [26,27].

## Swimming motility and chemotaxis promote persistence by enabling bacteria to counter intestinal flow and resist expulsion

We previously found that naturally aggregated bacteria are vulnerable to intestinal flow and
expulsion from the host [32,33]. To explore whether the attenuated colonization phenotypes
of Δmot and Δche are connected to their perturbed spatial organization in a way that causes
increased sensitivity to the intestine's mechanical forces, we followed the spatiotemporal
dynamics of wild-type *Vibrio* and each mutant in live animals by LSFM. Prior to imaging, each
strain was given 24 hours to reach its respective carrying capacity in germ-free zebrafish.
Despite wild-type *Vibrio* showing modest declines in abundance from 24 to 72 hpi (Fig 1A), it
was highly uniform and stable over periods of >10 hours, maintaining its abundance, low
cohesion, and foregut localization (Fig 3A, S3 Mov). We note that image-based quantification
of wild-type *Vibrio* abundances was performed in a concurrent study [33] and have been
replotted here. In contrast, Δmot and Δche underwent dramatic fluctuations in their abun-
dances and spatial organization (Fig 3A, S3 Mov). Cells and small aggregates in Δmot and
Δche populations appeared to become packed by intestinal contractions into large masses
within the midgut before being abruptly expelled. Autofluorescent material was often observed

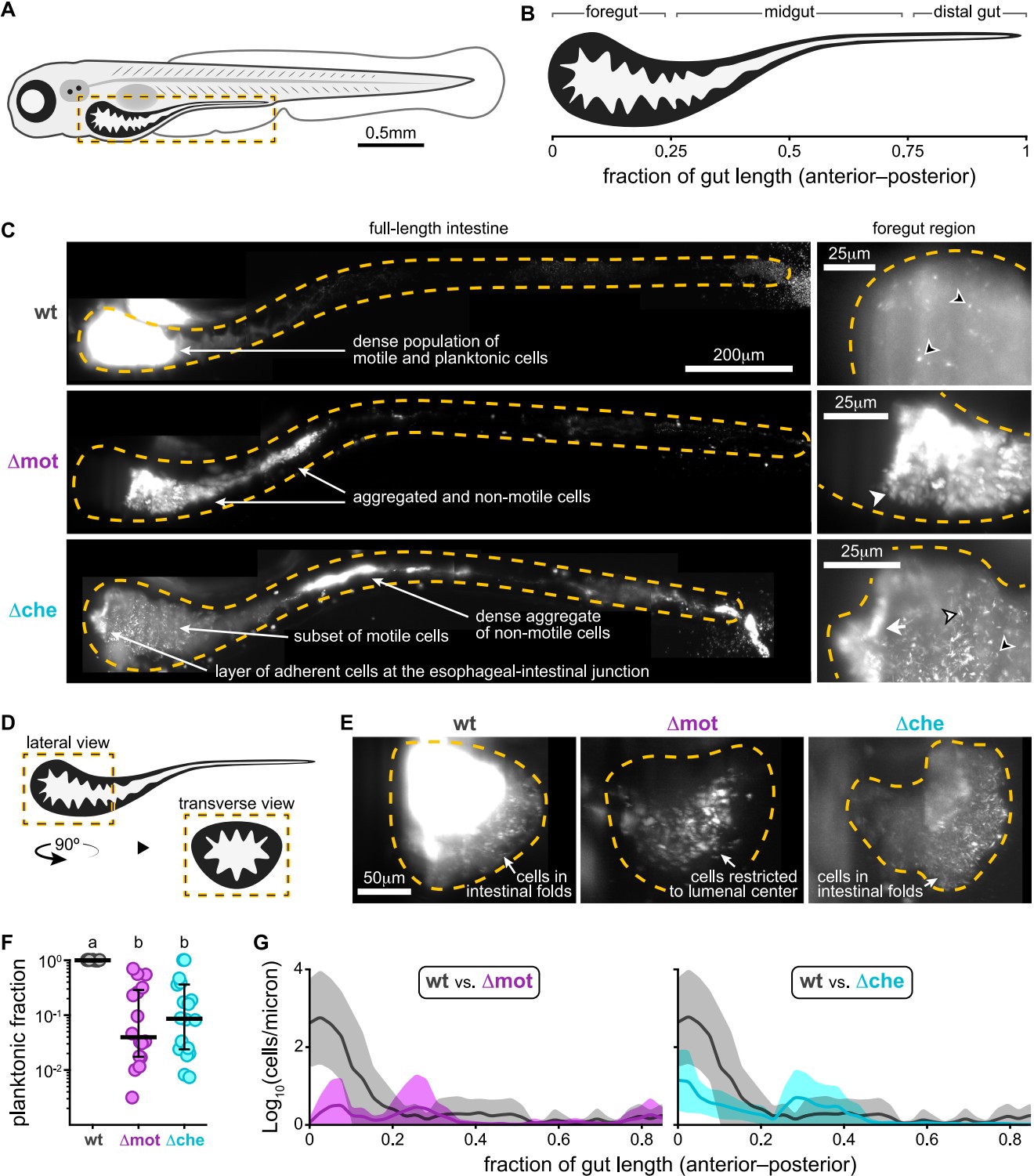

**Fig 2. Motility and chemotaxis mutants have altered intestinal spatial organization.** (A) Cartoon of a 6-day-old zebrafish. Dashed box marks intestinal region imaged by LSFM. (B) Anatomical regions of the larval zebrafish intestine. (C) Maximum intensity projections acquired by LSFM showing the spatial organization of wt *Vibrio* (top), Δmot (middle), and Δche (bottom) within the intestine. Top right inset shows a zoomed-in view of wt *Vibrio* cells in a separate fish that was colonized with a 1:100 mixture of green- and red-tagged variants so that the cellular organization of the dense *Vibrio* population could be discerned. The dilute channel (green) is shown. Dashed lines mark approximate intestinal boundaries. Open arrowheads: single bacterial cells; solid arrowheads: small aggregates; tailed arrowheads: large aggregates. Arrowheads with a black stroke mark swimming cells, which appear as comet-like streaks. (D) Cartoon showing the intestinal region pictured in panel E. (E) Maximum intensity projections acquired by LSFM showing transverse view of the

foregut region colonized with wt, Δmot, or Δche. (F) Fraction of planktonic cells contained within each strain's population. Each circle is a measurement from a single intestinal population. Bars denote medians and interquartile ranges. Letters denote significant differences. $p < 0.05$, Kruskal-Wallis and Dunn's multiple comparisons test. (G) Image-derived abundances of wt ($n = 7$), Δmot ($n = 4$), and Δche ($n = 5$) with respect to position along the length of the gut. Shaded regions mark confidence intervals. Underlying data plotted in panels F and G are provided in S1 Data. LSFM, light sheet fluorescence microscopy; wt, wild type.

surrounding aggregated cells, suggesting that host mucus was involved in this process. Image-based quantification of absolute abundances showed that >90% of Δmot and Δche populations could be lost in a single collapse event (Fig 3A). Following collapses, residual small aggregates in the midgut and low numbers of planktonic cells in the foregut appeared to undergo bursts in replication that effectively restored the abundance and spatial organization of the population before the next collapse. Notably, this pattern of aggregation, collapse, and regrowth mirrors other nonmotile symbiont populations, namely, those formed by zebrafish-derived *Aeromonas* and *Enterobacter* species [32,33]. Animating the relationship between cohesion and intestinal localization for each *Vibrio* strain across animals over time showed that both mutant populations exhibit large fluctuations in spatial organization whereas wt *Vibrio* populations are highly stable (S4 Mov).

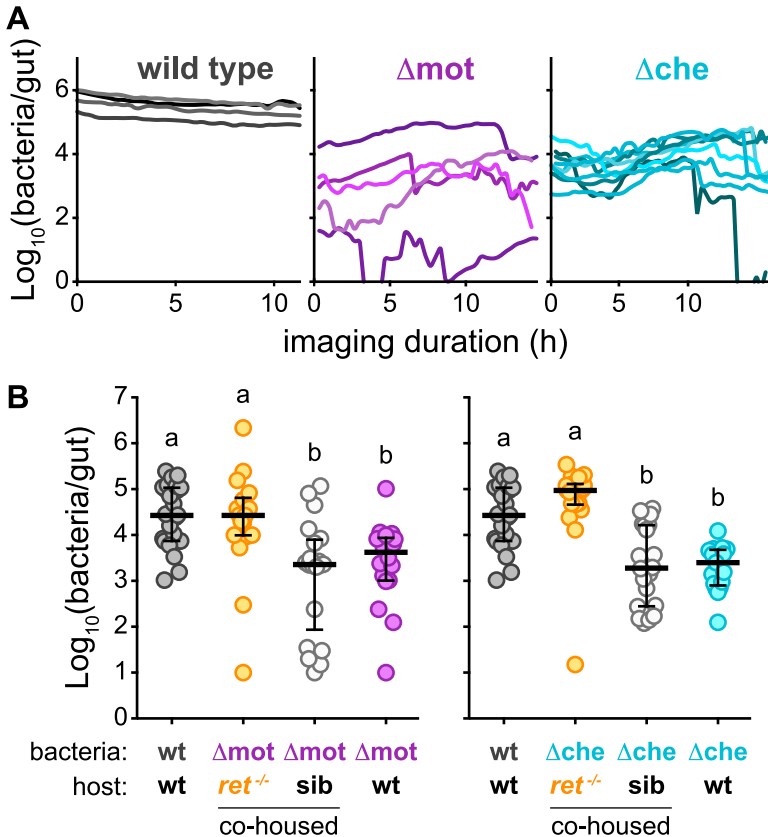

**Fig 3. Swimming motility and chemotaxis promote persistence by enabling bacteria to counter intestinal flow and resist expulsion.** (A) Image-based quantification of abundances over time for wt *Vibrio*, Δmot, and Δche. Lines represent individual populations in individual fish. (B) Cultivation-based quantification of abundances for Δmot and Δche in co-housed *ret*[−/−] mutant hosts and wt/heterozygous sib. Abundances of wt *Vibrio*, Δmot, and Δche in wt hosts (from Fig 1A, 72 hpi) are shown for comparison. Bars denote medians and interquartile ranges. Letters denote significant differences. $p < 0.05$, Kruskal-Wallis and Dunn's multiple comparisons test. Underlying data are provided in S1 Data. hpi, hours post inoculation; sib, sibling controls; wt, wild type.

Our live imaging results suggested that the altered spatial organization of Δmot and Δche populations, namely, their increased cohesion, makes them more susceptible to intestinal flow and expulsion and thus is likely the cause of their reduced abundances. This putative mechanism contrasts with the general assumption that swimming motility and chemotaxis primarily promote bacterial growth by facilitating nutrient foraging and avoidance of hostile environments. To probe the likelihood of these 2 different mechanisms, we quantified the in vivo exponential growth rates of Δmot and Δche (see Materials and methods). We found that both Δmot and Δche exhibit exponential growth rates within the intestine (Δmot = $0.7 \pm 0.3$ $h^{-1}$ [$n$ = 2]; Δche = $0.9 \pm 0.4$ $h^{-1}$ [$n$ = 5]) that are comparable to a previously determined wild-type *Vibrio* exponential growth rate of $0.8 \pm 0.3$ $h^{-1}$ (mean ± standard deviation) [32]. This result supports the idea that the reduced intestinal abundances of Δmot and Δche are not due to attenuated growth but rather are a consequence of altered behavior and spatial organization that increases susceptibility to intestinal flow and expulsion.

To test this expulsion-based mechanism more directly, we assessed whether the abundance of Δmot and Δche could be rescued in $ret^{-/-}$ mutant zebrafish hosts, which have reduced intestinal transport because of a dysfunctional enteric nervous system [32,42]. Humans with *ret* mutations can develop Hirschsprung disease, which is an affliction characterized by intestinal dysmotility and altered gut microbiome composition [43,44]. Strikingly, we found that the intestinal abundances of both Δmot and Δche were fully rescued to wild-type levels in $ret^{-/-}$ mutant animals (Fig 3B). In contrast, Δmot and Δche abundances in co-housed sibling control animals mirrored those in wild-type animals (Fig 3B). Importantly, we found that wild-type *Vibrio* shows no change in intestinal abundance in $ret^{-/-}$ mutant animals (S2A Fig), indicating that the rescue of Δmot and Δche is not due to a general overgrowth phenomenon. In addition, inspecting the spatial organization of Δmot in $ret^{-/-}$ mutant animals revealed that in some instances Δmot populations displayed relocalization to the anterior portion of the foregut, suggesting that intestinal flow is responsible for Δmot's posterior-shifted distribution in wild-type animals (S2B Fig). Together, these results provide further evidence that swimming motility and chemotaxis do not promote persistence by affecting growth per se but by enabling bacterial cells to physically resist intestinal flow and expulsion from the host.

## Sustained swimming motility is required for maintaining intestinal spatial organization and persistence

Without swimming motility, *Vibrio* has clear defects in both immigration and intestinal persistence. Therefore, we sought to experimentally separate the roles motility plays during these different stages of colonization. We specifically wanted to determine whether *Vibrio* requires sustained motility for intestinal persistence or if the impaired immigration and altered assembly of motility mutant populations was in some way responsible for their aggregated and collapsing phenotype. To accomplish this, we built a motility "loss-of-function" switch that uses inducible CRISPR interference (CRISPRi) to suppress transcription of the flagellar motor gene operon *pomAB* (Fig 4A and S3 Fig). The motility loss-of-function switch is based on a tetracycline induction system in which a constitutively expressed Tet repressor protein (TetR) is used to regulate the expression of a catalytically dead Cas9 (dCas9). We incorporated a constitutively expressed single-guide RNA (sgRNA) to target dCas9 to the 5' end of the native *pomAB* locus where it would block transcriptional elongation. To visually track switch activity in bacterial populations, we co-expressed *dcas9* with a gene encoding superfolder green fluorescent protein (sfGFP; Fig 4A). To mark all cells independent of switch activity, we co-expressed a gene encoding dTomato with *tetR*. Details on switch design and optimization are provided in Materials and methods and in S3A–S3D Fig. We integrated the motility loss-of-function

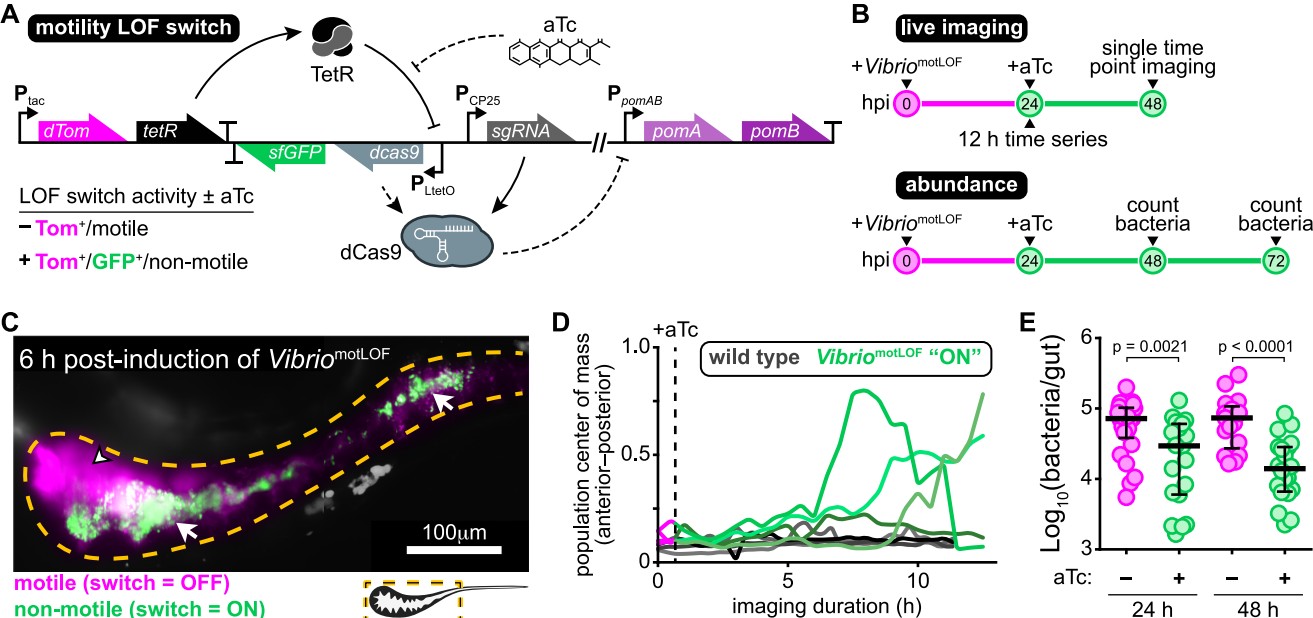

**Fig 4. Sustained swimming motility is required for maintaining intestinal spatial organization and persistence.** (A) Schematic of CRISPRi-based motility LOF switch. Lower left table summarizes switch activity and bacterial behaviors +/− aTc. Bent arrows denote promoters; "T" denotes transcriptional terminators. Solid lines represent constitutive interactions; dashed lines represent induced interactions. (B) Experimental timelines used to investigate in situ inactivation of swimming motility. (C) A maximum intensity projection acquired by LSFM of an animal colonized by *Vibrio*[motLOF] at 6 hpi. Dashed line marks approximate intestinal boundaries. An arrowhead with a black stroke marks an area of swimming cells expressing only dTomato (magenta, "switch = OFF"). White tailed arrowheads mark aggregated cells (green, "switch = ON"). (D) Population center of mass over time for intestinal populations of wild-type *Vibrio* (gray) and *Vibrio*[motLOF] (magenta/green). Lines are single bacterial populations within individual fish. Vertical dashed line marks time of aTc induction. (E) Abundances of *Vibrio*[motLOF] at 24 and 48 hpi with aTc. Bars denote medians and interquartile ranges. Significant differences determined by Mann-Whitney. Underlying data plotted in panels D and E are provided in S1 Data. aTc, anhydrotetracycline; CRISPRi, CRISPR interference; hpi, hours post induction; LOF, loss-of-function; LSFM, light sheet fluorescence microscopy.

switch into the genome of wild-type *Vibrio* (creating *Vibrio*[motLOF]) and confirmed that induction of the switch with the tetracycline analog anhydrotetracycline (aTc) robustly inactivates swimming motility in vitro without perturbing growth (S3E and S3F Fig).

With the motility loss-of-function switch constructed, we tested whether sustained swimming motility is required by established *Vibrio* populations to persist within the intestine using both live imaging and cultivation-based measurements of abundance (Fig 4B). For live imaging, germ-free zebrafish were first colonized to carrying capacity with *Vibrio*[motLOF]. At 24 hpi, repression of motility was induced by adding aTc to the water of colonized zebrafish hosts. We then performed time series imaging of multiple animals using LSFM. Initially, subpopulations emerged that could be distinguished by their switch activation status, behavior, and spatial organization (S5 Mov). Unswitched motile cells expressing only dTomato displayed a foregut localization pattern typical of wild-type *Vibrio* (Fig 4C). In contrast, we observed nonmotile cells expressing green fluorescent protein (GFP) becoming aggregated and segregating away from motile populations (Fig 4C). GFP-positive cells within aggregates were more restricted to the intestinal lumen and their arrangement suggested they were encased in mucus (Fig 4C and S5 Mov). By 10 h post induction, *Vibrio*[motLOF] displayed clear shifts in population center of mass toward the midgut together with expulsion of multicellular aggregates (Fig 4D).

Cultivation-based measures of absolute abundances revealed that at 24 h post induction *Vibrio*[motLOF] populations had an approximately 2.5-fold lower median abundance compared to uninduced controls (Fig 4E). Inducing for an additional 24 hours resulted in an approximately 5-fold reduction in median intestinal abundance (Fig 4E). Together, our experiments

using the motility loss-of-function switch demonstrate that *Vibrio* requires sustained swimming motility to maintain its spatial organization and to persist at high levels. Our results also reveal that relatively brief interruptions in *Vibrio*'s swimming behavior are capable of producing rapid and dynamic changes in spatial organization and drops in abundance.

## Acquisition of swimming motility or chemotaxis leads to rapid recovery of intestinal spatial organization and abundance

We next asked whether established Δmot and Δche populations could recover their spatial organization and abundance if they reacquired swimming motility or chemotaxis, respectively. Answering this question would give insight into the capacity of resident gut bacteria and would-be pathobionts to exploit a sudden loss of host spatial control. Using the motility loss-of-function switch backbone, we constructed motility and chemotaxis "gain-of-function" switches by inserting either *pomAB* or *cheA2* in place of *dcas9* (Fig 5A). The motility and chemotaxis gain-of-function switches were integrated into the genomes of Δmot and Δche, respectively, creating Δmot^GOF and Δche^GOF. In vitro tests showed that inducing the gain-of-function switches restored wild-type swimming behaviors in each strain without altering growth (S4A–S4C Fig). Moreover, activation of motility and chemotaxis prior to colonization produced intestinal abundances at 24 hpi that matched the carrying capacity of wild type (Fig 5B). These functional tests show that the motility and chemotaxis gain-of-function switches can be used to inducibly complement the Δmot and Δche mutants. Moreover, these genetic complementation experiments also show that the colonization phenotypes of Δmot and Δche were not due to off-target or polar effects resulting from our chromosomal manipulations.

We monitored the response dynamics of activating swimming motility or chemotaxis in established populations following similar experimental timelines as depicted in Fig 4B. Live imaging revealed that induced populations of Δmot^GOF and Δche^GOF underwent clear shifts in spatial distribution toward the foregut within the first 24 hours of induction compared to uninduced controls (Fig 5C). Strikingly, Δmot^GOF and Δche^GOF showed that large-scale changes in behavior and spatial organization could occur extremely rapidly, with both populations becoming more space-filling and foregut-localized within hours (Fig 5D, S6 and S7 Movs). Cultivation-based measurements of absolute abundances showed only modest increases in median intestinal abundances in the first 24 hours of induction (Fig 5E, 48 hpi). However, by 48 h post induction the median intestinal abundances of Δmot^GOF and Δche^GOF populations had recovered to wild-type levels (Fig 5E, 72 hpi). Therefore, regaining swimming behavior and undergoing spatial reorganization preceded the recovery of intestinal abundance.

Surprisingly, uninduced control populations of Δmot^GOF and Δche^GOF also exhibited a recovery in intestinal abundance by 72 hpi (S4D Fig). In vitro characterization and DNA sequencing revealed that this spontaneous recovery was likely due to nonsynonymous mutations in *tetR* that were acquired during intestinal colonization and impaired the function of the Tet repressor protein, thus resulting in constitutive switch activation. Although unexpected, we surmise that the extremely rapid sweep of "evolved clones" carrying disabled switches—which were rarely observed in induced populations or the aqueous environment outside the host (S4E Fig)—is evidence of strong selective pressures for motility traits within the gut.

## Motile bacterial cells within the intestine induce local and systemic *tnfa* expression

We next set out to connect *Vibrio*'s motility-based lifestyle to its pathogenic potential. We recently showed that overgrowth of *Vibrio*-related taxa sparks intestinal pathology marked by

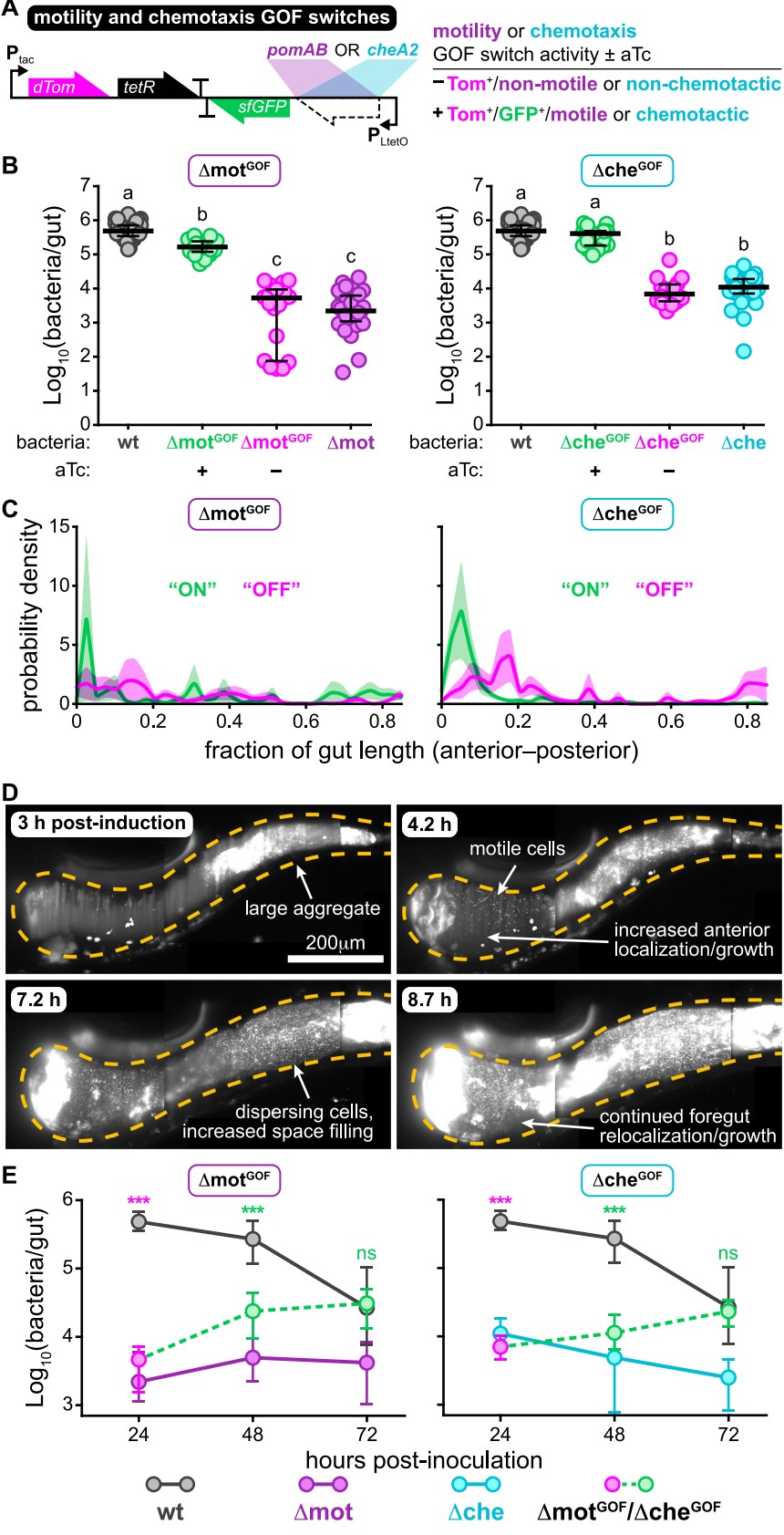

**Fig 5. Acquisition of swimming motility or chemotaxis leads to rapid recovery of intestinal spatial organization and abundance.** (A) Schematic of the motility and chemotaxis GOF switches. Table summarizes switch activity and bacterial behaviors +/− aTc. (B) $\Delta$mot$^{GOF}$ or $\Delta$che$^{GOF}$ abundances 24 hpi +/− aTc. $\Delta$mot$^{GOF}$ and $\Delta$che$^{GOF}$ were preinduced overnight in liquid culture prior to inoculation; aTc was maintained in the water for continuous switch activation. Abundances of wild-type *Vibrio*, $\Delta$mot, and $\Delta$che in wild-type hosts (from Fig 1A, 24 hpi) are shown for comparison. Bars denote medians and interquartile ranges. Letters denote significant differences. $p < 0.05$, Kruskal-Wallis and Dunn's multiple comparisons test. (C) Probability densities showing the spatial distributions of $\Delta$mot$^{GOF}$ and $\Delta$che$^{GOF}$ at 24 hpi. Magenta = uninduced; green = induced. Shaded regions mark standard errors. Sample sizes (populations within individual animals): $\Delta$mot$^{GOF}$ "OFF", $n = 5$; $\Delta$mot$^{GOF}$ "ON", $n = 7$, $\Delta$che$^{GOF}$ "OFF", $n = 6$; $\Delta$che$^{GOF}$ "ON", $n = 6$. (D) Maximum intensity projections acquired by LSFM from the same animal showing $\Delta$che$^{GOF}$ undergoing rapid changes in spatial organization following induction. Dashed lines mark approximate intestinal boundary. (E) Abundances of $\Delta$mot$^{GOF}$ and $\Delta$che$^{GOF}$ over time. Magenta and green circles indicate abundances +/− aTc, respectively. Plotted are medians and interquartile ranges ($n \geq 19$ animals/marker). Abundances of wild-type *Vibrio*, $\Delta$mot, and $\Delta$che (from Fig 1A) are shown for comparison. Significant differences between each mutant and wild-type determined by Mann-Whitney (magenta asterisks: uninduced; green asterisks: induced). $^{***}p < 0.0001$. Underlying data plotted in panels B, C, and E are provided in S1 Data. aTc, anhydrotetracycline; GOF, gain-of-function; hpi, hours post induction; LSFM, light sheet fluorescence microscopy; ns, not significant.

increased epithelial hypertrophy and neutrophil influx that is dependent on tumor necrosis factor alpha (TNFα) signaling [34]. We further identified that *Vibrio* ZWU0020 on its own can potently stimulate inflammation [36] and exacerbate pathology in susceptible hosts [34]. To explore the link between *Vibrio*'s motility behaviors and its inflammatory potential, we used LSFM and transgenic zebrafish hosts that express GFP under the control of the TNFα promoter (Tg(*tnfa*:GFP)) [45].

Germ-free animals displayed little *tnfa* reporter activity in or near the foregut where the bulk of wild-type *Vibrio* cells typically reside (Fig 6A). Similar to previous findings [45], animals colonized with a conventional, undefined microbial community also had low numbers of cells with *tnfa* reporter activity (Fig 6A). In contrast, at 24 hpi, wild-type *Vibrio* induced pronounced *tnfa* reporter activity in numerous host cells within both the intestine and liver (Fig 6A). All animals colonized with wild-type *Vibrio* had *tnfa*-expressing cells in or near the liver, whereas less than a third of germ-free and conventionalized animals had detectable *tnfa* reporter activity in this area (Fig 6B). Quantifying fluorescence intensity across the foregut region (including adjacent extraintestinal tissues and the liver) showed that *Vibrio* induces an approximately 100-fold increase in *tnfa* reporter activity over germ-free and conventional levels (Fig 6C). In contrast to wild-type *Vibrio*, $\Delta$mot and $\Delta$che elicited muted inflammatory responses. Animals colonized with $\Delta$mot showed a pattern of *tnfa* reporter activity similar to germ-free animals (Fig 6A–6C). However, despite the comparable intestinal abundances of $\Delta$mot and $\Delta$che (Fig 1A), $\Delta$che induced intermediate, although variable, levels of *tnfa* reporter activity (Fig 6A–6C). This finding suggests that host tissues do not merely sense bacterial abundances but also their active swimming behavior and/or proximity to epithelial surfaces. Together, these data provide evidence that swimming motility and chemotaxis are major contributors to *Vibrio*'s proinflammatory potential.

We next probed the possible host cell types involved in sensing motile *Vibrio* populations. The amoeboid morphology and migratory behavior of many *tnfa*-expressing cells hinted that they might be immune cells (Fig 6A and S8 Mov). Using double transgenic zebrafish carrying the *tnfa* reporter and expressing fluorescently marked macrophages (Tg(*mpeg1*:mCherry) [46]), we quantified the fraction of *tnfa*-positive macrophages within the foregut region, which we consider here as a field of view containing the foregut and adjacent tissues such as the liver. We found that approximately half ($54 \pm 10\%$ [mean $\pm$ standard deviation, $n = 100$ cells from 4 animals]) of the *tnfa*-positive cells in the foregut region induced by *Vibrio* were indeed macrophages (Fig 6D and S9 Mov). Nearly all *tnfa*-positive cells that were directly associated with the foregut were macrophages ($93 \pm 12\%$ [mean $\pm$ standard deviation, $n = 18$ cells from 3

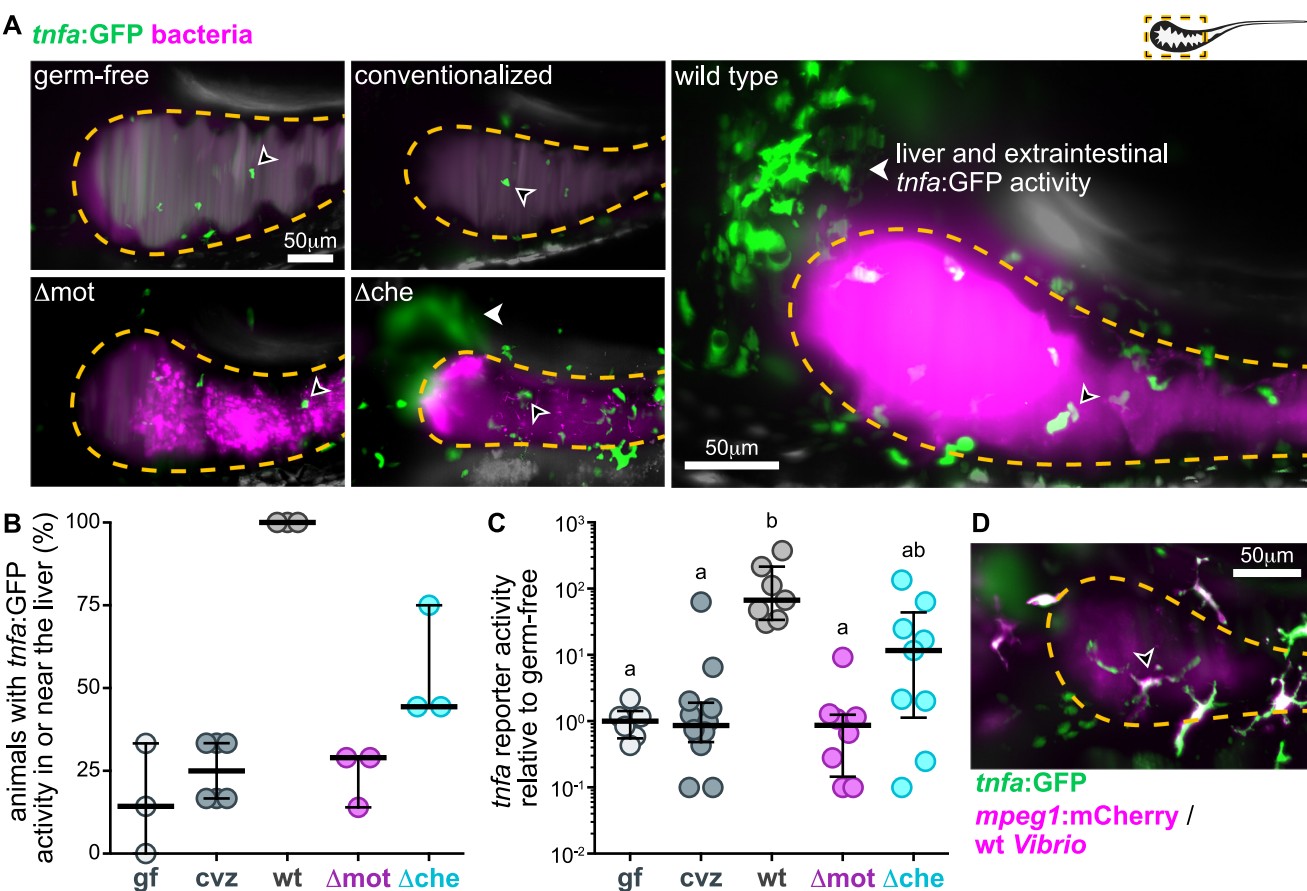

**Fig 6. Motile bacterial cells induce local and systemic *tnfa* expression.** (A) Maximum intensity projections acquired by LSFM of the foregut region of *tnfa*:GFP transgenic zebrafish raised germ-free, with a complex microbial community (conventionalized), or colonized solely with dTomato-expressing (magenta) wild-type *Vibrio*, Δmot, or Δche. Animals were imaged at 24 hpi. Dashed lines mark the approximate intestinal boundaries. Empty arrowheads mark host cells with *tnfa*:GFP reporter activity. Solid arrowheads mark *tnfa*:GFP reporter activity in extraintestinal tissues in or near the liver. (B) Percent of zebrafish subjected to different colonization regimes with *tnfa*:GFP activity in or near the liver; >6 animals/group were blindly scored by 3 researchers. Bars denote medians and interquartile ranges. (C) Total GFP fluorescence intensity across the foregut region normalized to median gf fluorescence intensity. Bars denote medians and interquartile ranges. Letters denote significant differences. $p < 0.05$, Kruskal-Wallis and Dunn's multiple comparisons test. (D) Maximum intensity projections acquired by LSFM of the foregut region of a *tnfa*:GFP, *mpeg1*:mCherry (magenta) transgenic zebrafish colonized with dTomato-expressing wild-type *Vibrio* (magenta). Animal was imaged at 24 hpi. Open arrowhead indicates a *tnfa*⁺/*mpeg1*⁺ cell. Underlying data plotted in panels B and C are provided in S1 Data. cvz, conventionalized; gf, germ-free; GFP, green fluorescent protein; hpi, hours post inoculation; LSFM, light sheet fluorescence microscopy.

animals]). In contrast, the majority of *tnfa*-positive cells associated within the liver did not appear to be macrophages based on *mpeg1*:mCherry expression, nor were they neutrophils (based on experiments with animals carrying an *mpx*:mCherry reporter), suggesting that they were other nonimmune cell types. Collectively, our data indicate that wild-type *Vibrio* populations stimulate expression of *tnfa* locally within intestinal tissues as well as at systemic sites, namely, the liver. Macrophages are also one of the main cell types that is sensitive to *Vibrio* colonization.

## Host tissues rapidly respond to sudden increases in bacterial swimming motility within the intestine

To maintain homeostasis, the host must be simultaneously tolerant and sensitive to the activity of resident bacterial populations. It is crucial for host tissues to quickly differentiate between

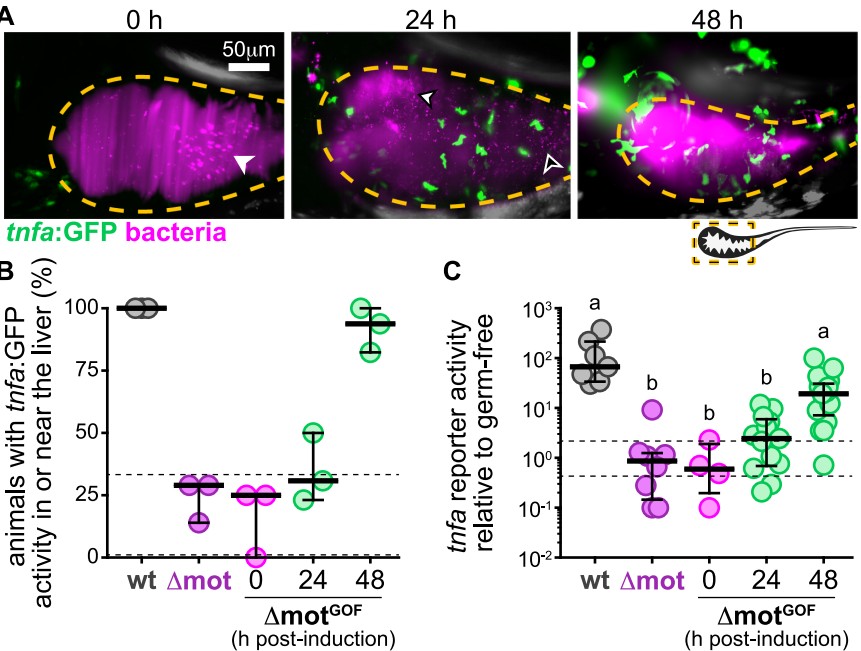

**Fig 7. Host tissues rapidly respond to sudden increases in bacterial swimming motility within the intestine.** (A) Maximum intensity projections acquired by LSFM of the foregut region of separate *tnfa*:GFP transgenic zebrafish colonized with ΔmotGOF (magenta). Dashed lines mark approximate intestinal boundaries. Times are hours post switch induction. Solid arrowhead marks bacterial aggregates, empty arrowhead marks single bacterial cells. (B) Percent of zebrafish subjected to different colonization regimes with *tnfa*:GFP activity in or near the liver; >4 animals/group were blindly scored by 3 researchers. Bars denote medians and interquartile ranges. Data from animals colonized with wt *Vibrio* or Δmot (from Fig 6B) are shown for comparison. Horizontal dashed lines mark gf range plotted in Fig 6B. (C) Total GFP fluorescence intensity across the foregut region normalized to median gf fluorescence intensity plotted in Fig 6C; horizontal dashed lines mark gf range. Bars denote medians and interquartile ranges. Data from animals colonized with wt *Vibrio* or Δmot (from Fig 6C) are shown for comparison. Letters denote significant differences. $p < 0.05$, Kruskal-Wallis and Dunn's multiple comparisons test. Underlying data plotted in panels B and C are provided in S1 Data. gf, germ-free; GFP, green fluorescent protein; LSFM, light sheet fluorescence microscopy; wt, wild type.

harmful and benign changes in the intestinal microbiota, for example, the overgrowth of a pathobiont versus diurnal fluctuations in commensal bacteria [47]. Therefore, we next determined if sudden increases in bacterial motility behaviors—which are a potential signature of pathobionts escaping host control—could elicit an equally rapid host response.

Following a similar live imaging timeline as depicted in Fig 4B, we used LSFM to track *tnfa* reporter activity in response to induced populations of ΔmotGOF. As expected, at time zero, ΔmotGOF populations displayed low abundance, high cohesion, and a posterior-shifted distribution with little *tnfa* reporter activity in host tissues (Fig 7A–7C). By 24 h post induction, ΔmotGOF populations had begun to spatially reorganize within the foregut and contained an increased number of swimming cells (Fig 7A). At the same time, there was an increase in *tnfa*-expressing host cells near the intestine, which were likely macrophages (Fig 7A). In one instance, we captured *tnfa*-positive host cells within the mucosa adjacent to bacterial cells actively swimming near the epithelial surface (S10 Mov). After the first 24 hours of induction, the fraction of animals with *tnfa*-postive cells in or near the liver did not increase appreciably (Fig 7B); however, there was an approximately 2.5-fold increase in median *tnfa* reporter activity, implying that initial responses to changes in bacterial swimming motility occur locally within intestinal tissues (Fig 7C). By 48 hours of switch induction, ΔmotGOF populations exhibited wild-type–like space-filling properties and foregut localization (Fig 7A). Likewise,

*tnfa* reporter activity was also mostly restored to wild-type levels (Fig 7B and 7C). Nearly all animals (approximately 92%, *n* = 17) had *tnfa*-positive cells in or near the liver and the median *tnfa* reporter activity across the foregut region was 20-fold higher than germ-free levels (Fig 7B and 7C). Our data reveal that host tissues are remarkably sensitive to sudden increases in bacterial motility behaviors that occur over relatively short time scales.

## Discussion

Our study connects the motile lifestyle of a gut bacterial symbiont to its colonization and proinflammatory potential. All gut bacteria must contend with host-mediated restrictions on microbiota spatial organization [48–50]. The mechanism by which *Vibrio* maintains stable colonization involves resisting intestinal flow through sustained swimming and chemotaxis. Conventional wisdom is that motility promotes the growth of bacteria by enabling them to forage nutrients and avoid hostile environments [14,17,18]. In contrast, our data show that *Vibrio*'s motility behaviors within the zebrafish gut do not enhance its exponential growth rate but rather allow it to resist intestinal expulsion. *Vibrio* thus provides a model of intestinal persistence that is distinct from more familiar examples involving adhesion to or invasion of host tissues, which are largely based on the examination of dissected and fixed samples and do not consider large-scale dynamics that play out across the entire gut [51–53]. Ultimately, *Vibrio*'s colonization strategy uses continuous swimming to remain in place within the host's intestine.

*Vibrio*'s swimming behavior underlies many of its pathobiont characteristics, including its ability to invade and displace resident bacteria, persist at high abundances, and stimulate host inflammation. The basis of *Vibrio*'s inflammatory activity is presently unknown. On the host side, our work implicates macrophages as a host cell type that is capable of responding to *Vibrio*'s motile behavior through the up-regulation of TNFα, but whether the mechanism of sensing *Vibrio* is direct or indirect remains to be determined. We also observed other host cells at systemic sites, particularly within the liver, that up-regulate TNFα in response to wild-type *Vibrio* colonization. Transcriptional profiling (e.g., via single-cell RNA sequencing) and transgenic animals carrying genetic reporters will be useful for probing the identity of these additional host cell types as well as the receptor(s) and signaling cascades involved in sensing *Vibrio* populations.

On the bacterial side, future work is aimed at testing how host inflammation is connected to *Vibrio*'s abundance, position along the intestine, mucosal proximity, and cellular behavior. Intriguingly, the different inflammatory activities of Δmot and Δche, despite their similar spatial organizations and production of flagella, highlights the possibility that the mechanism involves active bacterial motility. We posit that motility allows bacteria to access epithelial surfaces, increasing concentrations of inflammatory molecules at host cell surfaces and possibly triggering mechanosensory pathways. For example, flagellar rotation itself has been shown to increase the shedding of immunogenic lipopolysaccharide and outer membrane vesicles in other *Vibrio* lineages [54,55]. It is also possible that the Δmot and Δche mutants differ from wild-type in their level of expression of inflammatory flagellar components despite displaying intact flagella in vitro.

*Vibrio*'s colonization dynamics show how swimming motility and chemotaxis enable gut bacteria to evade spatial constraints imposed by the host. Notably, Δmot and Δche reveal how bacteria with impaired motility surrender to intestinal mechanics, which act to confine them within the lumen where they can be periodically purged. The aggregation of Δmot and Δche within the intestine was unexpected, especially for Δche, which displays vigorous swimming in vitro. It is possible that both mutants experience shifts in metabolism or induce biofilm behaviors in vivo as a consequence of their inability to effectively control their spatial distribution. It

is also possible that the aggregation of Δmot and Δche stems from a defect in flagellar assembly or signaling, which could be disentangled using mutants lacking flagella altogether. Alternatively, Δmot and Δche cells may become entrapped within intestinal mucus and grow locally to produce small clonal aggregates that are subsequently consolidated into larger aggregates by intestinal mechanics before being collectively expelled. Corroborating this idea, it was shown in an infant mouse model that the attenuated colonization of a human-derived isolate of *V. cholerae* lacking motility can largely be reversed in animals that are pretreated with a mucolytic agent that disrupts mucus architecture [56]. Furthermore, it is possible that the swimming activity of Δche may actually facilitate mucus entrapment. It has been shown that nonchemotactic, straight-swimming bacterial cells that are unable to periodically redirect their swimming trajectory can become jammed within a porous medium (such as in a soft agar matrix commonly used to study chemotaxis in vitro) [57]. We propose that a similar mechanism could lead to entrapment and aggregation of Δche cells within intestinal mucus, which wild-type *Vibrio* cells avoid because they are able to actively escape from mucus through regular changes in swimming direction mediated by chemotactic signaling.

More broadly, our study supports the idea that in addition to the intestine's role in transporting digesta and expelling waste, intestinal flow and mucus dynamics also appear to exert spatial and population control over nonmotile and nonchemotactic resident microbiota. Consistent with this, our previous characterization of *sox10* mutant zebrafish demonstrated how the enteric nervous system can prevent microbiome-mediated inflammation and pathology by constraining intestinal bacterial abundances and composition [34]. Further, we recently showed that intestinal mechanics can amplify the impact of sublethal antibiotic treatment on gut bacteria, which induces bacterial aggregation and thus leads to enhanced intestinal expulsion [33]. A similar phenomenon was described in the mouse intestine, where antibody-mediated enchaining of bacterial cells enhanced clearance of *S.* Typhimurium [58]. Our observation that intestinal flow impacts the distribution of bacteria throughout the gut is also corroborated by findings in "gut-on-a-chip" fluidic systems [59].

Given the clear advantage of motility behaviors within the gut, it is somewhat surprising that the majority of zebrafish gut bacteria studied so far—many of which are capable of flagellar motility—form aggregated populations made up of mostly nonmotile cells [26]. This discrepancy may be reconciled by considering the broader ecological life cycles of gut bacteria. For example, we have found that intestinal populations of *Aeromonas* grow more rapidly within multicellular aggregates than as planktonic cells [60]. Moreover, *Aeromonas* also benefits from swimming motility during interhost dispersal [37]. *Aeromonas* thus highlights how aggregation and expulsion by intestinal flow may actually facilitate growth and transmission in the context of a population of hosts [61]. Bacterial aggregation may also be part of a bacterial strategy for preventing host inflammation and avoiding subsequent antimicrobial responses. Supporting this idea, we recently found that several aggregated *Aeromonas* species are sensitive to host inflammation [62]. In contrast, the particular *Vibrio* strain used in the present work is largely tolerant to host inflammation [36] and thus can stimulate host inflammatory responses without consequence.

Further investigation of the relationship between intestinal mechanics and gut bacterial lifestyles will open new avenues for therapeutic engineering of the gut microbiome. Our findings suggest that manipulating bacterial motility and aggregation may be used to induce large-scale, yet specific, changes in both bacterial abundances and host inflammatory state. Moreover, using drug- or diet-based modulators of intestinal flow may enhance the efficacy of antibiotics or promote microbiome recovery and fortification following perturbation. Our experiments using genetic switches to toggle bacterial motility and inflammatory activity serve as a proof-of-concept for these types of manipulations. Highlighting the potential of these

interventions, human studies have shown that colonic transit time is a top predictor of microbiome composition [63,64]. Moreover, impaired intestinal flow can lead to bacterial overgrowth and pathogenic changes in the microbiome [34,44,65]. Considering the dynamic nature of the intestinal ecosystem on spatial and temporal scales relevant to bacterial cells will be key to therapeutically engineering the microbiome.

## Materials and methods

### Ethics statement

All experiments with zebrafish were done in accordance with protocols approved by the University of Oregon Institutional Animal Care and Use Committee and following standard protocols (protocol number 15–98) [66]. Specific handling and housing of animals during experiments are described in detail under the section "Gnotobiology". All zebrafish used in this study were larvae, between the ages of 4- and 7-days post fertilization. Sex differentiation occurs later in zebrafish development and thus was not a factor in our experiments.

### Zebrafish lines

Zebrafish lines used in this study included: University of Oregon stock wild-type ABCxTU; zebrafish carrying the $ret1^{hu2846}$ mutant allele [32,42]; zebrafish carrying the Tg(*tnfa*:GFP) transgene [45]; and zebrafish carrying the Tg(*mpeg1*:mCherry) transgene [46]. Double transgenic animals included Tg(*tnfa*:GFP) x Tg(*mpeg1*:mCherry) and Tg(*tnfa*:GFP) x Tg(*mpx*:mCherry) [67]. Of note, $ret1^{hu2846}$ is recessive and adult zebrafish carrying this mutant allele were maintained as heterozygotes. Incrossing $ret1^{hu2846}$ animals produces $ret^{+/+}$, $ret^{+/-}$, and $ret^{-/-}$ individuals. $ret^{-/-}$ larvae can be visually distinguished from $ret^{+/+}$ and $ret^{+/-}$ larvae based on developmental features. In our study we classified $ret^{+/+}$ and $ret^{+/-}$ larvae together as "sibling" controls.

### Bacterial strains and culture

**General.** All wild-type and recombinant bacterial strains used or created in this study are listed in S1 Table. Archived stocks of bacteria are maintained in 25% glycerol at −80˚C. Prior to manipulations or experiments, bacteria were directly inoculated into 5 mL lysogeny broth (10 g/L NaCl, 5 g/L yeast extract, 12 g/L tryptone, 1 g/L glucose) and grown for approximately 16 hours (overnight) shaking at 30˚C. For growth on solid media, tryptic soy agar was used. Gentamicin (10 μg/mL; Amresco, Solon, OH) was used to select recombinant *Vibrio* strains during their creation (for both gene deletion and insertion variants). Ampicillin (100 μg/mL; Gold Biotechnology, St. Louis, MO) was used for maintaining plasmids in *E. coli* strains.

**In vitro growth measurements.** In vitro growth of bacterial strains was assessed using the FLUOstar Omega microplate reader (BMG LABTECH, Offenburg, Germany). Prior to growth measurements, bacteria were grown overnight in 5 mL lysogeny broth at 30˚C with shaking. The next day, cultures were diluted 1:100 into fresh lysogeny broth and dispensed in triplicate or quadruplicate (i.e., 3–4 technical replicates; 200 μl/ well) into a sterile 96-well clear flat bottom tissue culture-treated microplate (Corning, Corning, NY). Anhydrotetracycline (aTc; Takara Bio USA, Mountain View, CA). Absorbance measurements at 600 nm were recorded every 30 minutes for 16 hours (or until stationary phase) at 30˚C with shaking. Growth measurements were repeated at least 2 independent times for each strain (i.e., 2 biological replicates) with consistent results. Data plotted are from a single replicate.

**In vitro motility assays.** The swimming behavior of each *Vibrio* strain was assessed using soft agar assays and live imaging of bacterial motility in liquid media on glass slides. For soft agar assays, bacteria were first grown overnight in 5 mL lysogeny broth at 30˚C with shaking.

One milliliter of bacterial culture was then washed by centrifuging cells at 7,000$g$ for 2 minutes, aspirating media and suspending in 1 mL 0.7% NaCl. This centrifugation and aspiration wash step was repeated once more, and bacteria were suspended in a final volume of 1 mL 0.7% NaCl. One microliter of washed bacterial cells was inoculated into swim agar plates made of tryptic soy agar containing 0.2% agar. In the case of Δmot$^{GOF}$ and Δche$^{GOF}$, aTc was also added to the agar at the indicated concentrations. Swim plates were incubated at 30˚C for 6 hours and imaged using a Gel Doc XR+ Imaging System (Bio-Rad, Hercules, CA). For live imaging of swimming behavior, bacteria were first grown overnight in 5 mL lysogeny broth at 30˚C with shaking. The next day, cultures of wild-type *Vibrio*, Δmot, and Δche were diluted 1:100 in tryptic soy broth and grown for 2 hours with shaking at 30˚C. Δmot$^{GOF}$ and Δche$^{GOF}$ were diluted 1:100 in tryptic soy broth ±50 ng/mL aTc and grown for 4 hours with shaking at 30˚C. *Vibrio*$^{motLOF}$ was diluted 1:1000 in tryptic soy broth ±50 ng/mL aTc and grown for 7 hours with shaking at 30˚C. Prior to imaging, bacteria were diluted 1:40 in tryptic soy broth, mounted on glass slides with a coverslip and imaged for 10 seconds using a Nikon Eclipse Ti inverted microscope equipped with an Andor iXon3 888 camera. Representative maximum intensity projections of 10 second movies shown in S1, S3 and S4 Figs were generated in FIJI [68]. For measurements of swimming behavior, bacteria were tracked using the radial center algorithm [69] for object localization and nearest-neighbor linking. Motility assays were repeated at least 2 independent times (i.e., 2 biological replicates) with consistent results.

**Scanning electron microscopy.** Bacteria were prepared for environmental scanning electron microscopy (ESEM) by first growing cells on tryptic soy agar overnight at 30˚C. A sterile inoculating loop was used to transfer approximately 100 μl of cells to a 1.6 mL tube containing 500 μl of 3% glutaraldehyde fixative (Sigma-Aldrich, St. Louis, MO). We visually confirmed that *Vibrio* cells isolated from an agar plate are highly motile and thus capable of producing flagella during culture on solid media. Cells were fixed overnight at 4˚C. The next day, cells were sequentially washed in increasing concentrations of ethanol: first in plain ddH$_2$O followed by 20%, 40%, 60%, and 80% ethanol. Each wash involved centrifuging cells at 7,000$g$ for 2 minutes, aspirating media, and suspending in the next wash medium. A small aliquot of washed cell suspension was applied to a silicon wafer, dried, and imaged using a FEI Quanta 200 ESEM/VPSEM environmental scanning electron microscope provided by the University of Oregon's Center for Advanced Materials Characterization in Oregon (CAMCOR) facility.

**Disk diffusion assays.** Disk diffusion assays were often used to test and optimize dTomato and sfGFP reporter function of genetic switches as described in S3C and S3D Fig. Bacteria were first grown overnight in 5 mL lysogeny broth at 30˚C with shaking. One hundred microliters of dense overnight culture were spread onto tryptic soy agar plates to produce a lawn of growth. Prototyping was typically done using plasmid-base switches in *E. coli* (as was the case in S3C and S3D Fig), thus tryptic soy agar plates also contained ampicillin to ensure plasmid maintenance. A sterile piece of Whatman filter paper (approximately 0.5 cm wide) was placed in the center of the plate and impregnated with approximately 2 μg of aTc. Plates were incubated overnight at 30˚C. Switch reporter activity was assessed using a Leica MZ10 F fluorescence stereomicroscope equipped with 1.0×, 1.6×, and 2.0× objectives and a Leica DFC365 FX camera (Leica, Wetzlar, Germany). Images were captured and processed using standard Leica Application Suite software and FIJI [68].

## Molecular techniques and genetic manipulations

**General.** Recombinant strains used or created in this study are listed in S1 Table. Plasmids used or created in this study are listed in S2 Table. Primer and oligo DNA sequences are listed in S3 Table.

*E. coli* strains used for molecular cloning and conjugation were typically grown in 5 mL lysogeny broth at 30°C or 37°C with shaking in the presence of appropriate antibiotic selection to maintain plasmids. For propagation of *E. coli* on solid media, LB agar was used. Unless specified, standard molecular techniques were applied, and reagents were used according to manufacturer's instructions. Restriction enzymes and other molecular biology reagents for polymerase chain reaction (PCR) and nucleic acid modifications were obtained from New England BioLabs (Ipswich, MA). Various kits for plasmid and PCR amplicon purification were obtained from Zymo Research (Irvine, CA). The Promega Wizard Genomic DNA Purification Kit was used for isolating bacterial genomic DNA (Promega, Madison, WI). DNA oligonucleotides were synthesized by Integrated DNA Technologies (IDT; Coralville, IA). Sanger sequencing was done by Sequetech (Mountain View, CA) to verify the sequence of all cloned genetic parts. A Leica MZ10 F fluorescence stereomicroscope with 1.0×, 1.6×, and 2.0× objectives and Leica DFC365 FX camera were used for screening fluorescent bacterial colonies.

Genome and gene sequences were retrieved from "The Integrated Microbial Genomes & Microbiome Samples" (IMG/M) website (https://img.jgi.doe.gov/m/) [70]. Where applicable, "IMG" locus tags are provided for genetic loci, which can be used to access sequence information via the IMG/M website.

**Construction of gene deletions.**   Markerless, in-frame gene deletions were constructed using allelic exchange and the pAX1 allelic exchange vector (Addgene Plasmid #117397) as previously described [26]. Detailed procedures and protocols can be accessed online: https://doi.org/10.6084/m9.figshare.7040264.v1. Creation of Δmot via deletion of *pomAB* (locus tags: ZWU0020_01568 and ZWU0020_01567) (S1A Fig) was reported previously [26]. Creation of Δche via deletion of *cheA2* (locus tag: ZWU0020_00514) (S1A Fig) was accomplished by first constructing a *cheA2* allelic exchange cassette using splice by overlap extension (SOE). The *cheA2* allelic exchange cassette was designed to fuse the start and stop codons of the *cheA2* gene (S1A Fig). PCR primer pairs WP165 + WP166 and WP167 + WP168 were used to amplify 5' and 3' homology regions flanking the *cheA2* gene, respectively, from *Vibrio* ZWU0020 genomic DNA. The resulting amplicons were spliced together, and the SOE product was ligated into a pAX1-based allelic exchange vector, producing pAX1-ZWU0020-cheA2 (pTW383). After subsequent subcloning steps, the final sizes of the 5' and 3' homology regions were 763 bp and 845 bp.

The pAX1-ZWU0020-cheA2 vector was delivered into *Vibrio* via conjugation (i.e., bacterial mating) as previously described using *E. coli* SM10 as a donor strain [26]. Briefly, *Vibrio* and SM10/pAX1-ZWU0020-cheA2 were combined 1:1 on a filter disk (EMD Millipore, Billerica, MA) placed on tryptic soy agar. The mating mixture was incubated at 30°C overnight. Following incubation, bacteria were recovered and spread onto tryptic soy agar containing gentamicin and incubated overnight at 37°C to select for *Vibrio* merodiploids. Merodiploid colonies were isolated and screened for successful deletion of the *cheA2* gene. Putative mutants were genotyped by PCR using primers that flanked the *cheA2* locus (WP0169 + CheA2.ZW20. KOconfirm.REV), which produced 2 differently sized amplicons representing the wild-type and mutant alleles (S1A Fig).

**Design and construction of genetic switches.**   Customizable, plasmid-based gain-of-function (pXS-GOF-switch, pTW265) and loss-of-function (pXS-LOF-switch, pTW308) switch scaffolds were initially constructed and optimized using the pXS-dTomato (Addgene Plasmid #117387) backbone, which was previously generated [26]. The general architecture of switch elements is depicted in S3A Fig. Each element is flanked by unique restriction sites to allow straightforward insertion of new elements by restriction cloning. pXS-dTomato contains the "tracker" element, which comprises a constitutive $P_{tac}$ promoter (without the lac operator sequence) [26] driving the *dTomato* gene. The "switch reporter" element was first inserted,

which comprises a $P_{LtetO}$ promoter [71] driving a *sfGFP* gene that was amplified from pTW168 using WP138 + WP118. Next, the "repressor" element was inserted, which comprises a *tetR* gene that was amplified from *Enterobacter* ZOR0014 genomic DNA using WP146 + WP139. As described in S3B and S3C Fig, a near-random ribosome binding site (ndrrdn) was incorporated by PCR into the 5' untranslated region of the *tetR* gene via WP146. A clone containing the ribosome binding site sequence "ctaggt" was isolated that had strong reporter repression/induction and robust tracker expression. Next, as described in S3B and S3D Fig, a ribozyme-based insulator sequence (RiboJ) [72] was inserted between the switch reporter and the insertion site designated to hold switch "cargo" genes. The RiboJ sequence was inserted using a custom synthesized gBlock gene fragment (IDT). The resulting plasmid-based switch scaffold—comprising a tracker, switch reporter, repressor, and RiboJ sequence—became pXS-GOF-switch. To generate pXS-LOF-switch, we inserted the *dcas9* gene [73] (excised from pdCAS9, Addgene Plasmid #44249) as the "cargo" element and a constitutively expressed single-guide RNA ("sgRNA" element) driven by the CP25 promoter [74], which was inserted using a custom synthesized gBlock gene fragment (IDT). The stock sgRNA that was inserted into the pXS-LOF-switch is based on a previously characterized sgRNA specific for the *lacZ* gene of *E. coli* [73], which facilitated optimization of loss-of-function switch activity in *E. coli* K-12 (MG1655). To expedite insertion of the gain-of-function and loss-of-function switches into the *Vibrio* chromosome, each switch scaffold was subcloned into the previously described Tn*7* delivery vector pTn7xTS (Addgene Plasmid #117389), creating pTn7xTS-GOF-switch (pTW285) and pTn7xTS-LOF-switch (pTW317). We note that insertion of the switch scaffolds into the pTn7xTS vector limits some downstream customization due to restriction site conflicts.

To construct the motility loss-of-function switch, the *lacZ* sgRNA in the pTn7xTS-LOF-switch was replaced with a sgRNA specific for the *Vibrio pomA* gene, creating pTn7xTS-mot-LOF-switch (pTW340). The *pomA* sgRNA was inserted using a custom synthesized gBlock gene fragment (IDT). To construct the motility gain-of-function switch, the *pomAB* locus, including the native *pomA* ribosome binding site, was amplified using WP170 + WP171 and inserted into the cargo site of pTn7xTS-GOF-switch, creating pTn7xTS-mot-GOF-switch (pTW324). To construct the chemotaxis gain-of-function switch, the *cheA2* locus, including the native *cheA2* ribosome binding site, was amplified using WP92 + WP93 and inserted into the cargo site of pTn7xTS-GOF-switch, creating pTn*7*xTS-che-GOF-switch (pTW282).

**Tn7-mediated chromosomal insertions.** Chromosomal insertion of fluorescent markers and genetic switches was done via a Tn*7* transposon-based approach using the Tn*7* delivery vector pTn7xTS as previously described [26]. Detailed procedures and protocols can be accessed online: https://doi.org/10.6084/m9.figshare.7040258.v1. Specific pTn7xTS vectors carrying markers or switches were delivered into *Vibrio* via triparental mating using 2 *E. coli* SM10 donor strains carrying either the pTn7xTS delivery vector or the pTNS2 helper plasmid (Addgene Plasmid #64968). Briefly, *Vibrio* and SM10 donor strains were combined 1:1:1 on a filter disk placed on tryptic soy agar. The mating mixture was incubated at 30°C overnight. Following incubation, bacteria were recovered and spread onto tryptic soy agar containing gentamicin and incubated overnight at 37°C to select for *Vibrio* insertion variants. Insertion of the Tn*7* transposon and the genetic cargo it carried into the *attTn7* site near the *glmS* locus of *Vibrio* was confirmed by PCR using primers WP11 + WP12.

Fluorescently marked wild-type *Vibrio* constitutively expressing dTomato (ZWU0020 *attTn7::dTomato*) was previously generated using pTn7xTS-dTomato (Addgene Plasmid #117391) [26]. In the current work, fluorescently marked Δmot and Δche were constructed in the same way, creating Δmot *attTn7::dTomato* and Δche *attTn7::dTomato*. *Vibrio*^motLOF^ was created by inserting the motility loss-of-function switch from pTn7xTS-mot-LOF-switch.

$\Delta$mot$^{GOF}$ was created by inserting the motility gain-of-function switch from pTn7xTS-mot-GOF-switch. $\Delta$che$^{GOF}$ was created by inserting the chemotaxis gain-of-function switch from pTn7xTS-che-GOF-switch.

## Gnotobiology

**Germ-free derivation.** For all experiments, zebrafish embryos were initially derived germ-free using previously described gnotobiotic procedures with slight modification [75]. Briefly, fertilized eggs from adult mating pairs were harvested and incubated in sterile embryo media (EM) containing ampicillin (100 μg/mL), gentamicin (10 μg/mL), amphotericin B (250 ng/mL; MP Biomedicals, Santa Ana, CA), tetracycline (1 μg/mL; Sigma-Aldrich, St. Louis, MO), and chloramphenicol (1 μg/mL; Amresco, Solon, OH) for approximately 6 hours. Embryos were then washed in EM containing 0.1% polyvinylpyrrolidone-iodine (Syndel, Ferndale, WA) followed by EM containing 0.003% sodium hypochlorite (Fisher Scientific, Hampton, NH). Surface sterilized embryos were distributed into T25 tissue culture flasks (TPP, Trasadingen, Switzerland) containing 15 mL sterile EM at a density of 1 embryo/mL and kept in a temperature-controlled room at 28˚C to 30˚C with a 14 hours/ 10 hours light/ dark cycle. The germ-free status of larval zebrafish was assessed before every experiment by visually inspecting flask water for microbial contaminants using an inverted microscope. Culture-based assessment of germ-free status was done as needed by plating 100 μL flask water on rich media (e.g., tryptic soy agar). Embryos were sustained on yolk-derived nutrients and not fed prior to or during any experiments.

**Bacterial associations.** For bacterial associations, bacterial strains were grown overnight in lysogeny broth with shaking at 30˚C and prepared for inoculation by pelleting the cells from 1 mL of culture for 2 minutes at 7,000*g* and washed once in sterile EM. For all experiments, except where noted otherwise, washed bacteria were inoculated into the water of T25 flasks containing 4-day-old larval zebrafish at a final density of approximately 10$^6$ bacteria/mL. For competition experiments, *Vibrio* strains were added to the water of *Aeromonas*-colonized zebrafish (at 5 days old) without removing the original *Aeromonas* inoculum from the water. In addition, to enable enumeration of *Aeromonas* and *Vibrio* strains on agar plates, competition experiments were done using a previously constructed dTomato-expressing *Aeromonas* strain (*Aeromonas attTn7::dTomato*) [26]. For LOF and GOF switch experiments involving cultivation-based abundance measurements, prior to aTc-induction zebrafish were washed and placed in sterile EM to ensure that changes in intestinal populations were not interfered with by bacteria in the water. To conventionalize animals (i.e., colonize with a complex, undefined microbial consortium), 0- and 4-day-old larval zebrafish were inoculated with 100 μL of water taken from parental spawning tanks. No difference was found between conventionalization times in terms of host *tnfa*:GFP expression.

## Cultivation-based measurement of abundances

Dissection of larval zebrafish guts was done as previously described with slight modification [76]. Briefly, dissected guts of tricaine-euthanized zebrafish were harvested and placed in a 1.6 mL tube containing 500 μL sterile 0.7% saline and 100 μL 0.5 mm zirconium oxide beads (Next Advance, Averill Park, NY). Guts were homogenized using a bullet blender tissue homogenizer (Next Advance, Averill Park, NY) for 25 seconds on power 4. Lysates were serially plated on tryptic soy agar and incubated overnight at 30˚C prior to enumeration of colony forming units and determination of bacterial abundances. Abundance data presented throughout the main text and in S2 Fig are pooled from a minimum of 2 independent experiments (*n* = 16–36 dissected guts per condition). Abundance data presented for $\Delta$mot$^{GOF}$ and $\Delta$che$^{GOF}$

without aTc induction in S4 Fig are from a single representative experiment ($n = 8$–$10$ dissected guts per condition; water abundances are from single measurements). Samples with zero countable colonies on the lowest dilution were set to the limit of detection (5 bacteria per gut). Data were plotted and analyzed using GraphPad Prism 6 software. Unless stated otherwise, statistical differences between 2 groups of data were determined by Mann-Whitney; statistical differences between 2 paired groups of data were determined by Wilcoxon; and statistical differences among 3 or more groups of data were determined by Kruskal-Wallis test with Dunn's multiple comparisons test.

## Live imaging

**LSFM.** Live larval zebrafish were imaged using a custom-built light sheet fluorescence microscope previously described in detail [60]. Prior to mounting, larvae were anesthetized with MS-222 (also known as tricaine; Syndel, Ferndale, WA). A metal plunger was used to mount fish into small glass capillaries containing 0.5% agarose gel. Samples were then suspended vertically, head up, in a custom imaging chamber containing EM and anesthetic. Larvae in the set gel were extruded from the end of the capillary and oriented such that the fish's left side faces the imaging objective. For experiments involving just fluorescent bacteria, the approximately 1 mm long intestine is imaged in 4 subregions that are registered in software after imaging. A single 3D image of the full intestine volume (approximately $200 \times 200 \times 1{,}200$ microns) sampled at 1-micron steps between z-planes is imaged in approximately 45 seconds. For experiments including the *tnfa*:GFP reporter, only one subregion containing the anterior foregut region and approximately 100 microns of tissue anterior to the gut was captured for the majority of samples. In these image stacks, nearly the full extent of the fish's left–right width was captured, approximately 400 microns in z. For time lapse imaging of genetic switch induction, fish were mounted as normal and baseline dynamics were captured for 30 to 90 minutes depending on the experiment. Then, the inducer aTc was added to the sample chamber media in an approximately 1 mL solution of EM, MS-222, and aTc. Excitation lasers of wavelengths 488 and 561 nm were adjusted to a power of 5 mW as measured before the imaging chamber. An exposure time of 30 ms was used for all 3D scans and 2D movies. Time lapse imaging was performed overnight, except for the additional growth rate measurement for Δche, which occurred during the day. For color images presented within figures, autofluorescent tissues—namely, the yolk, swim bladder, and ventral skin—were manually converted to grayscale to enhance clarity.

**Identification of fluorescent bacteria.** Identification of bacteria in zebrafish images was conducted using a previously described computation pipeline written in MATLAB [27,60]. In brief, individual bacteria are first identified with a combination of wavelet filtering [77], standard difference of Gaussians filtering, intensity thresholding, and manual curation. Then, multicellular aggregates, which are too dense to resolve individual cells, are segmented via a graph cut algorithm [78] seeded with an intensity mask. The number of cells per multicellular aggregate is estimated by dividing the total aggregate fluorescence intensity by the mean intensity of single cells. These estimates of number of cells per bacterial object in the gut are then used to compute spatial distributions along the length of the gut, following a manually drawn line drawn that defines the gut's center axis.

**Measurement of planktonic fractions.** Planktonic fractions were computed for each fish by dividing the number of identified single cells by the total abundance. For wild-type *Vibrio*, the primary population of motile bacteria is typically too dense for us to resolve single cells. However, sparse labeling experiments, in which fish were colonized with GFP- and dTomato-marked *Vibrio* at a ratio of 1:100 (Fig 2C, top row, right), indicated that this population was

indeed entirely planktonic. Movies showing motility of sparsely labeled cells were previously published [26,27]. Therefore, to compute planktonic fraction for wild-type *Vibrio* in single-labeled populations, we treated the dense, anterior subpopulation computationally just like we would an aggregate—using our aggregate detection algorithm and estimating the number of cells present by normalizing the total fluorescence intensity to the mean single-cell intensity—but then counted these cells as planktonic.

**Measurement of in vivo growth rates.** Through time lapse imaging and the computational image analysis methods discussed above, bacterial growth rates in the intestine can be directly measured by linear fits to log-transformed abundances [32,33,60]. The in vivo growth rate of wild-type *Vibrio* was previously measured [32]. The in vivo growth rate for Δmot was measured in the time traces shown in Fig 3A, using manually defined windows of clear exponential growth. To exclude effects of density dependence on the growth rate, only those traces that began at least 1 order of magnitude below the median Δmot abundance were considered. For Δche, the time traces in Fig 3A were insufficient for growth rate estimation, because abundances remained at high levels for most of the experiment. Therefore, we measured the growth rate in populations shortly after initial colonization. Specifically, Δche was allowed to colonize germ-free fish for 6 hours, after which fish were mounted for time lapse imaging. Previous work on another zebrafish gut bacterial symbiont showed that exponential growth rates in established and nascent populations are equal [32]. Abundance data for these time traces are included in the S1 Data.

**Quantification of *tnfa*:GFP fluorescence.** Cells and tissues expressing *tnfa*:GFP were segmented in 3D with basic intensity threshold-based segmentation. A pixel intensity threshold of 1,500 was empirically found to be a conservative threshold and was used for all samples. The 488 nm excitation laser power was set at 5 mW prior to entering the sample chamber for all samples. The camera was a pco.edge scientific CMOS camera (PCO, Kelheim, Germany). The resulting identified objects were then filtered by size to remove noise. GFP signal from near the ventral skin was excluded with a manually defined cropped region created in the ImageJ software [68]. Green autofluorescence from the interior gut region rarely passed the intensity and size thresholds to contribute to measured *tnfa*:GFP signal. Similarly, in the motility GOF switch experiments, we found that the GFP reporter of switch induction never reached fluorescence intensity levels high enough to contribute measurably to the *tnfa*:GFP signal. Nevertheless, this region was automatically identified and removed via intensity threshold-based segmentation in the red 568/620 nm (excitation/emission) color channel. Both red autofluorescence and signal from red (dTomato) fluorescent bacteria were used to identify this gut region. Finally, to standardize total *tnfa*:GFP quantification across different samples, an operational peri-intestinal region was defined as containing the foregut plus all tissue 100 microns anterior of the start of the gut, which was automatically identified in the mask generated from the red color channel. Automatic gut segmentation and removal was not performed for the dual *tnfa*:GFP/*mpeg1*:mCherry reporter fish.

**Measuring *tnfa*:GFP/*mpeg1*:mCherry fluorescence.** Red fluorescence from *mpeg1*:mCherry marking macrophages was segmented analogously to *tnfa*:GFP signal, using basic intensity threshold-based segmentation in 3D and size filtering. A *tnfa*+ object and an *mpeg*+ object were considered to overlap if their centroids were separated by less than 10 microns, a threshold empirically determined to produce accurate results as judged by eye. The fraction of *tnfa*+ objects that were also *mpeg*+ and the fraction of *mpeg*+ objects that were also *tnfa*+ were computed using the counts for overlapping and nonoverlapping cells.

**Data and statistical analysis.** Data were plotted using MATLAB and GraphPad Prism 6 software. Statistical analyses were done using GraphPad Prism 6. Unless stated otherwise, medians and interquartile ranges were plotted. Statistical tests performed are specified in

figure legends and within Materials and methods under "Cultivation-based measurement of abundances". $p \leq 0.05$ was considered significant for all analyses. Sample sizes are noted within the main text, figure legends, within Materials and methods under "Cultivation-based measurement of abundances", and in S1 Data (.xls). What "n" represents is specified in the main text and figure legends.

## Supporting information

**S1 Fig. Motility and chemotaxis mutant construction and in vitro characterization.** (A) Gene diagrams depict the in-frame markerless deletion of *pomAB* (Δmot construction) and *cheA2* (Δche construction). "Δ" denotes the mutant allele and the DNA sequence shown below represents the resulting fusion of the start and stop codons in each case. Black triangles represent primers used for PCR-confirmation of each mutant, and the amplicon sizes (bp) of the wt and mutant (Δ) alleles are provided above each locus. DNA gels to the right of each diagram show the successful deletion of both *pomAB* and *cheA2* from the *Vibrio* chromosome. We note that the Δmot mutant was constructed in a previous publication [26], and the DNA gel shown is a version of that already published but is included here for continuity and thoroughness. ns: nonspecific amplicon. (B, Left) Swimming motility of wt, Δmot, and Δche in liquid media and soft agar. Motility in liquid media was recorded for 10 seconds on a glass slide. Images show cellular movements over the entire 10 second period, which illustrates each cell's swimming trajectory. Swim distances were captured 6 hpi of bacteria into the agar. (Right) Probability densities showing the distribution of cellular swimming speeds in liquid media for each *Vibrio* strain. Sample sizes (measured bacterial swim tracks): wt = 2,962; Δmot = 754; Δche = 3,069. Inspection of cultures before and during analyses provided no obvious indications of elevated in vitro aggregation for Δmot or Δche compared to wt. (C) In vitro growth curves of each *Vibrio* strain in rich media (lysogeny broth). Line traces the average OD from 4 replicate wells; bars indicate range. Notably, smooth growth curves reiterate that Δmot or Δche do not exhibit elevated aggregation in vitro. (D) Scanning electron micrographs of each *Vibrio* strain after growth on solid media. Images show that each strain is capable of assembling a single polar flagellum. Underlying data plotted in panels B and C are provided in S1 Data. bp, base pair; hpi, hours post inoculation; OD, optical density; PCR, polymerase chain reaction; wt, wild type. (EPS)

**S2 Fig. Additional wild-type and Δmot colonization data in *ret*$^{-/-}$ mutant hosts.** (A) Cultivation-based abundances for wild-type *Vibrio* in co-housed *ret*$^{-/-}$ mutant hosts and wild-type/heterozygous sib. Abundances of wild-type *Vibrio* in wild-type hosts (from Fig 1A, 72 hpi) are shown for comparison. Letters denote significant differences. $p < 0.05$, Kruskal-Wallis and Dunn's multiple comparisons test. (B) Maximum intensity projections acquired by LSFM from a sib host (top) or a *ret*$^{-/-}$ mutant host (bottom). Each animal was colonized with Δmot for 72 hours prior to imaging. Dashed lines mark approximate intestinal boundaries. Underlying data plotted in panel A are provided in S1 Data. hpi, hours post inoculation; LSFM, light sheet fluorescence microscopy; sib, sibling controls. (EPS)

**S3 Fig. Switch design features and in vitro characterization of the motility loss-of-function switch.** (A) Diagram depicts the customizable design of the switch scaffold. Unique rs flanking switch elements allow each component to be optimized or replaced. The "tracker" encodes a fluorescent protein for marking all bacterial cells. The "repressor" encodes a transcription factor that allows inducible control of "cargo" gene (e.g., *dcas9*) expression. The "switch reporter" encodes a fluorescent protein that is co-expressed with the cargo gene to signal switch

activation. A "sgRNA" is inserted when the switch is used for CRISPRi. (B) Gene diagram indicates the locations (cyan bullseyes) of 2 elements that were essential for switch function: an optimized *tetR* RBS and a ribozyme-based insulator. (C, Left) Shown are DNA sequences for the native (top) and functionally optimized (bottom) 5' UTR of the *tetR* gene. Underlined cyan text denotes the RBS. Bolded text marks the *tetR* start codon. The middle sequence represents the library of *tetR* 5' UTRs containing randomized RBS sequences that were screened (letters are based on IUPAC code). (Right) Switch function was assessed using disk diffusion assays in which *E. coli* carrying the switch (without a cargo gene inserted) were spread at a density high enough to produce a lawn of growth on an agar plate. A disk impregnated with concentrated aTc was then used to induce switch activity, thereby making the adjacent cells express GFP if the switch was functional. (Top right) Original switch prototypes failed to be induced and displayed suppressed expression of the dTomato tracker, which we surmised was due to overexpression of TetR. (Bottom right) A library of switch clones containing random RBS sequences in the *tetR* 5' UTR were screened, resulting in the recovery of a functional clone that displayed sensitive switch activation and robust tracker expression. (D, Top row) Early switch prototypes relied on the co-transcription of the cargo gene and sfGFP reporter. However, the insertion of large cargo genes, such as a *dcas9* or *cheA2*, hampered sfGFP expression compared to an "empty" switch without a cargo gene, which was evident in disk diffusion assays. We surmised that this was due to part-junction interference between *sfGFP* and the cargo, leading to poor translation of *sfGFP*. (Bottom row) Insertion of the self-cleaving RiboJ ribozyme insulator between *sfGFP* and the cargo alleviated the apparent interference. (E, Left) Swimming motility of ΔmotLOF in liquid media ± aTc (50 ng/mL). Motility in liquid media was recorded for 10 seconds on a glass slide. Images show cellular movements over the entire 10 second period, which illustrates each cell's swimming trajectory. Motility was assessed approximately 7 h post-induction. (Middle) The percentage of swimming cells in ΔmotLOF populations in liquid media ± aTc (50 ng/mL) from 4 separate fields of view. (Right) Probability densities showing the distribution of cellular swimming speeds in liquid media for ΔmotLOF ± aTc (50 ng/mL). Sample sizes (measured bacterial swim tracks): ΔmotLOF − aTc = 2,677; ΔmotLOF + aTc = 944. (F) In vitro growth curves of ΔmotLOF in rich media (lysogeny broth) ± aTc (50 ng/mL). Line traces the average OD from 3 replicate wells; bars indicate range. Underlying data plotted in panels E and F are provided in S1 Data. aTc, anhydrotetracycline; CRISPRi, CRISPR interference; GFP, green fluorescent protein; hpi, hours post induction; OD, ocular density; RBS, ribosome binding site; rs, restriction sites; sfGFP, superfolder green fluorescent protein; sgRNA, single-guide RNA; TetR, Tet repressor protein; UTR, untranslated region.
(EPS)

**S4 Fig. In vitro characterization of motility and chemotaxis GOF switches and supporting data on the evolution of GOF switches in vivo.** (A, Left) Swimming motility of ΔmotGOF and ΔcheGOF in liquid media ± aTc (50 ng/mL). Motility in liquid media was recorded for 10 seconds on a glass slide. Images show cellular movements over the entire 10 second period, which illustrates each cell's swimming trajectory. Motility was assessed approximately 4 h post induction. (Right) Motility of wild-type *Vibrio*, ΔmotGOF, and ΔcheGOF in soft agar ± aTc (10 ng/mL). Swim distances were captured 6 hpi of bacteria into the agar. (B) Swim distances of wild-type *Vibrio*, ΔmotGOF, and ΔcheGOF in soft agar 6 h post induction with different concentrations of aTc. (C) In vitro growth curves of ΔmotGOF and ΔcheGOF in rich media (lysogeny broth) ± aTc (50 ng/mL). Line traces the average OD from 3 replicate wells; bars indicate range. (D) Cultivation-based abundances of ΔmotGOF or ΔcheGOF at 72 hpi either with (green) or without (magenta) aTc induction. Abundances of wild-type *Vibrio* (gray), Δmot (purple), and Δche (cyan) (from Fig 1A, 72 hpi) are shown for comparison. Abundances of each GOF

strain in the presence of aTc are from Fig 5E and are also shown for comparison. Bars denote medians and interquartile ranges. Letters denote significant differences. $p < 0.05$, Kruskal-Wallis and Dunn's multiple comparisons test. (E) Shown is the fraction of "evolved clones" (i.e., bacterial colonies recovered that displayed constitutive switch activation) from the intestines of zebrafish colonized with $\Delta$mot$^{GOF}$ or $\Delta$che$^{GOF}$ at 72 hpi that were either induced (green) or not induced (magenta) with aTc. Bars denote medians and interquartile ranges. In each condition, a black dashed bar and diamond labeled with a "w" indicates the fraction of evolved clones recovered from the water environment. Underlying data plotted in panels B–E provided in S1 Data. aTc, anhydrotetracycline; GOF, gain-of-function; hpi, hours post inoculation; OD, optical density.
(EPS)

**S1 Mov. Montage of real time movies showing wild-type *Vibrio*, Δmot, and Δche within larval zebrafish intestines.** Movies were acquired by LSFM at 24 hpi. Wild-type *Vibrio* is highly motile and planktonic, with swimming cells frequently making close contact with the intestinal epithelium. The bright signal in the left side of the frame is a mass of motile cells that is too dense for individuals to be resolved (see Fig 2C). In contrast, Δmot is largely aggregated and confined to the lumen. The Δche mutant exhibits an intermediate phenotype consisting of a motile subpopulation that is less dense than wild-type populations. The first 3 fields of view shown center on the foregut region. The fourth field of view ("che-midgut") centers on the midgut to highlight aggregates of Δche cells within this region. For the fourth field of view, intensities were log-transformed to highlight both the structure of the aggregates and the motile, planktonic cells visible in the lower left. Scale bar = 50 μm. hpi, hours post inoculation; LSFM, light sheet fluorescence microscopy.
(MP4)

**S2 Mov. Montage of animated z-stacks showing wild-type *Vibrio*, Δmot, and Δche within larval zebrafish intestines.** Movies were acquired by LSFM at 24 hpi. Wild-type *Vibrio* is highly motile and planktonic, with swimming cells frequently making close contact with the intestinal epithelium. The bright signal in the left side of the frame is a mass of motile cells that is too dense for individuals to be resolved (see Fig 2C). In contrast, Δmot is largely aggregated and confined to the lumen. The Δche mutant exhibits an intermediate phenotype consisting of a motile subpopulation that is less dense than wild-type populations. The field of view centers on the foregut region. The label in the upper right corner denotes the depth in z (left–right) through the intestine. Scale bar = 50 μm. hpi, hours post inoculation; LSFM, light sheet fluorescence microscopy.
(MP4)

**S3 Mov. Montage of time lapse movies showing wild-type *Vibrio*, Δmot, and Δche within larval zebrafish intestines.** Movies were acquired by LSFM starting at approximately 24 hpi. Wild-type *Vibrio* cells, which are highly motile and planktonic, robustly localize to the foregut region. The bright signal in the left side of the frame is a stable mass of motile cells that is too dense for individuals to be resolved (see Fig 2C). In contrast, Δmot is largely aggregated, confined to the lumen, and exhibits large fluctuations in spatial organization, including the rapid expulsion of a large aggregate. The Δche mutant exhibits an intermediate phenotype, consisting of a motile subpopulation that is less dense than wild-type populations with large, multicellular aggregates. A large aggregate of Δche cells is expelled near the end of the movie. The field of view spans the entire larval intestine. Scale bar = 200 μm. hpi, hours post inoculation; LSFM, light sheet fluorescence microscopy.
(MP4)

**S4 Mov. Animation of the spatiotemporal dynamics of wild-type *Vibrio*, Δmot, and Δche within larval zebrafish intestines.** Through computational image analysis, bacterial populations were segmented and enumerated. From this quantification, we computed the fraction of the population that were single cells (planktonic fraction) and computed the population center of mass along the length of the gut (population center). Each marker represents an entire bacterial population from an individual fish. The movie depicts the time evolution of multiple populations in this 2D phase space. Wild-type *Vibrio* populations robustly localize to the foregut region and maintain a high planktonic fraction. In contrast, Δmot and Δche populations undergo large fluctuations in aggregation and localization over time. (MP4)

**S5 Mov. Inactivation of swimming motility in established *Vibrio*^motLOF populations using the motility LOF switch.** Shown are 2 examples of *Vibrio*^motLOF switching dynamics within the larval zebrafish intestine captured by LSFM. *Vibrio*^motLOF initially colonized each intestine in a phenotypically wild-type state (i.e., switch = "OFF") with cells expressing only dTomato (magenta) and displaying a strong localization to the foregut and a high fraction of motile cells. At time zero, populations were induced by addition of aTc to the media. Both examples show the emergence of a multicellular aggregate from the anterior population of motile cells, a posterior-shift in overall distribution, and an increase in GFP expression signaling switch activation. Scale bars for time-lapses are 200 μm. Each frame of the time lapses are maximum intensity projections of a 3D image stack across the full intestinal volume. For the second example, we highlight the 3D structure of an emerging bacterial aggregate (arrow) with an animated rendering (dTomato fluorescence only). Scale bar for the rendering is 50 μm. The montage ends with a real time movie of *Vibrio*^motLOF cells approximately 16 h post induction showing widespread loss of motility (dTomato fluorescence only). Real time movie scale bar = 50 μm. aTc, anhydrotetracycline; GFP, green fluorescent protein; hpi, hours post induction; LOF, loss-of-function, LSFM, light sheet fluorescence microscopy. (MP4)

**S6 Mov. Activation of swimming motility in an established Δmot^GOF population using the motility GOF switch.** Δmot^GOF initially colonized the intestine with the motility GOF switch in the "OFF" state and therefore was nonmotile and assembled a population that was aggregated and had a poster-shifted distribution. At time zero, the population was induced by addition of aTc to the media. The resulting switching dynamics were captured by LSFM. Each frame of the time lapse is a maximum intensity projection of a 3D image stack across the full intestinal volume. Over time, motile cells appear and occupy the foregut region. Scale bar = 200 μm. Following the time lapse, we show a real time movie of a different fish at approximately 6 h post induction that captures induced Δmot^GOF cells swimming within the foregut. Real time movie scale bar = 50 μm. aTc, anhydrotetracycline; GOF, gain-of-function; hpi, hours post induction; LSFM, light sheet fluorescence microscopy. (MP4)

**S7 Mov. Activation of chemotaxis in an established Δche^GOF population using the chemotaxis GOF switch.** Δche^GOF initially colonized the gut with the chemotaxis GOF switch in the "OFF" state and therefore was nonchemotactic and assembled a population that was aggregated and had a poster-shifted distribution. At time zero, the population was induced by addition of aTc to the media. The resulting switching dynamics were captured by LSFM. Each frame of the time lapse is a maximum intensity projection of a 3D image stack across the full intestinal volume. Over time, there is a dramatic increase in the number of planktonic and motile cells that occupy the foregut region. Scale bar = 200 μm. Following the time lapse, we

show a real time movie of a different fish at approximately 6 h post induction that captures induced $\Delta$che$^{GOF}$ cells swimming within the foregut. Real time movie scale bar = 50 μm. aTc, anhydrotetracycline; GOF, gain-of-function; hpi, hours post induction; LSFM, light sheet fluorescence microscopy.
(MP4)

**S8 Mov. Migratory behavior of *tnfa*:GFP$^+$ cells.** Time lapse movie of a live *tnfa*:GFP transgenic larval zebrafish showing the migratory behavior of gut-associated *tnfa*$^+$ cells (arrowheads). Images were acquired by LSFM. Each frame of the time lapse is a maximum intensity projection of a 3D image stack that captures the full intestinal volume. Scale bar = 200 μm. GFP, green fluorescent protein; LSFM, light sheet fluorescence microscopy.
(MP4)

**S9 Mov. Animated z-stack of a *tnfa*:GFP/*mpeg1*:mCherry double transgenic larval zebrafish colonized with wild-type *Vibrio*.** The *mpeg1*:mCherry reporter and *Vibrio* dTomato marker were imaged simultaneously using a single excitation and emission system and are shown in magenta. *tnfa*:GFP fluorescence is shown in green. Images were acquired by LSFM. We first show an animated z-stack that depicts single planes of the light sheet with the depth (left–right) indicated in the upper right. *tnfa*$^+$ and *mpeg1*$^+$ single-positive cells, as well as *tnfa*$^+$/*mpeg1*$^+$ double-positive cells, are apparent. Scale bar = 50 μm. Following the animated z-stack, we show a 2-color, 3D rendering. Rendering scale bar = 50 μm. GFP, green fluorescent protein; LSFM, light sheet fluorescence microscopy.
(MP4)

**S10 Mov. Spatial distribution of *tnfa*$^+$ host cells responding to swimming bacterial cells within the intestine.** Montage shows the foregut region of a larval zebrafish carrying the *tnfa*:GFP reporter (green) colonized with $\Delta$mot$^{GOF}$ (magenta) 24 h post induction of the motility GOF switch with aTc. Images and real time movie were acquired by LSFM. We first show an animated z-stack that depicts single planes of the light sheet with the depth (left–right) indicated in the upper right. Arrows indicate *tnfa*$^+$ host cells and bacterial cells. Next, we show a 3D rendering of the same intestine, which highlights the association of *tnfa*$^+$ host cells with the mucosa. The montage ends with a real time movie of a single optical plane showing the swimming behavior of induced $\Delta$mot$^{GOF}$ cells relative to *tnfa*$^+$ host cells within the mucosa. All scale bars = 50 μm. aTc, ; GFP, green fluorescent protein; GOF, gain-of-function; hpi, hours post induction; LSFM, light sheet fluorescence microscopy.
(MP4)

**S1 Table. Bacteria used and created in this study.**
(PDF)

**S2 Table. Plasmids used and created in this study.**
(PDF)

**S3 Table. Primer and oligo DNA sequences.**
(PDF)

**S1 Data. File (.xls) containing all plotted numerical data.**
(XLSX)

**S1 Raw images. File containing images of DNA gels shown in S1 Fig.**
(PDF)

## Acknowledgments

We thank Maria Bañuelos and other members of the Guillemin lab as well as Dr. Judith Eisen and members of her lab for thoughtful discussion throughout this study. We also thank Rose Sockol and the University of Oregon Zebrafish Facility staff for fish husbandry and care.

## Author Contributions

**Conceptualization:** Travis J. Wiles, Brandon H. Schlomann, Raghuveer Parthasarathy, Karen Guillemin.

**Formal analysis:** Travis J. Wiles, Brandon H. Schlomann.

**Funding acquisition:** Travis J. Wiles, Brandon H. Schlomann, Raghuveer Parthasarathy, Karen Guillemin.

**Investigation:** Travis J. Wiles, Brandon H. Schlomann, Elena S. Wall.

**Methodology:** Travis J. Wiles, Brandon H. Schlomann, Elena S. Wall, Reina Betancourt.

**Supervision:** Raghuveer Parthasarathy, Karen Guillemin.

**Visualization:** Travis J. Wiles, Brandon H. Schlomann.

**Writing – original draft:** Travis J. Wiles, Brandon H. Schlomann.

**Writing – review & editing:** Travis J. Wiles, Brandon H. Schlomann, Elena S. Wall, Raghuveer Parthasarathy, Karen Guillemin.

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
