## [Editor Report · Decision Letter 0]

9 Dec 2019

Dear Dr Guillemin, 

Thank you for submitting your manuscript entitled "Swimming motility of a gut bacterial symbiont promotes resistance to intestinal expulsion and enhances inflammation" for consideration as a Research Article by PLOS Biology.

Your manuscript has now been evaluated by the PLOS Biology editorial staff as well as by an academic editor with relevant expertise and I am writing to let you know that we would like to send your submission out for external peer review.

Please re-submit your manuscript within two working days, i.e. by Dec 11 2019 11:59PM.

Kind regards,

Lauren A Richardson, Ph.D

Senior Editor

PLOS Biology

---

## [Decision Letter · Decision Letter 1]

22 Jan 2020

Dear Dr Guillemin,

Thank you very much for submitting your manuscript "Swimming motility of a gut bacterial symbiont promotes resistance to intestinal expulsion and enhances inflammation" for consideration as a Research Article by PLOS Biology. As with all papers reviewed by the journal, yours was evaluated by the PLOS Biology editors as well as by an Academic Editor with relevant expertise and by independent reviewers. 

As you will read, the reviewers all found work very well done and insightful. Based on the reviews, we are very pleased to let you know that we will probably accept this manuscript for publication, assuming that you will modify the manuscript to address the remaining points raised by the reviewers. Please note that no additional experimental work is required and make sure to address the data and other policy-related requests noted at the end of this email. 

We expect to receive your revised manuscript within two weeks. Your revisions should address the specific points made by each reviewer. In addition to the remaining revisions and before we will be able to formally accept your manuscript and consider it "in press", we also need to ensure that your article conforms to our guidelines. A member of our team will be in touch shortly with a set of requests. As we can't proceed until these requirements are met, your swift response will help prevent delays to publication.

*Copyediting*

*Published Peer Review History*

*Early Version*

*Submitting Your Revision*

Sincerely,

Lauren A Richardson, Ph.D

Senior Editor

PLOS Biology

DATA POLICY:

**I have assessed the data you have provided and find it complete. My only request is to remove the sentences "Underlying raw image data are available from the authors by request. DNA sequences, genetic constructs, and engineered bacteria are available from the authors by request." We do not allow "by request" availability and the data you have provided if sufficient to meet our requirements. You may keep the section in your manuscript "Data and code availability" as is, but please remove these sentences from the official Data Availability Statement.

Reviews

Reviewer #1: Will Ludington, signed review

Wiles et al investigate the persistence of a natural pathobiont bacteria in the larval zebrafish gut and the consequences for immune activation in the fish. They develop and use several genetic tools to precisely turn bacterial motility on and off. They demonstrate a multiscale mechanism tying genes in the bacteria and host to physical flows in the fish gut. They also detail several feedbacks between the physical and physiological parts of the system. Overall, loss of bacterial motility induces bacterial aggregation and makes them susceptible to host-generated clearing forces. The authors also investigate the host immune response, including its spatial dynamics in response to bacterial motility. The results have implications for understanding microbiome-host dynamics in many systems including humans. This is one of the most comprehensive works I have read recently and it gives deep mechanistic insights into the host microbiome relationship. I think it will have major impacts on the field because it gives us a systems level understanding in mechanistic detail. Few microbiome studies have linked genetics to physical forces, and the authors demonstrate that the zebrafish system that they have developed is particularly powerful for this type of work.

I have minor comments to help the authors improve the manuscript, but, in general, I think it can be published as is. I do not think any additional experiments need to be done for this manuscropt to be published in PLOS Biology. I would like the authors to address my comments through very minor clarifications to the writing.

line 91 "spacing-filling"  "space-filling"

line 150 It is puzzling that the motility defect also slows entry of cells into the intestine. Is there a hypothesis for why? Maybe this could be addressed in the discussion?

line 190 Good summary. Please give a little more indication of the hypothesis for how chemotaxis affects in vivo function of Vibrio. Is it simply that ∆che cells can swim but don't?

line 299-300: What is the growth rate as a function of the bacterial abundance? Growth rates could increase in populations that are turning over faster. 

line 317 Similar comment to line 190: please hypothesize why chemotaxis blocks aggregation -- or is it just a loss of motility? 

Fig 5C,D. Why is there such a high aggregation at the anterior end? Are those adherent? I would expect more even distribution for motile cells such as in Fig. 6A wt cells.

line 437 Nice control and follow up. This rapid evolution story is an interesting result hidden in this already substantial body of work.

Do the ∆mot and ∆che strains produce normal levels of flagellin on a per cell basis? Flagella appear normal in S1, but do all cells look that way? Could you do a Western? NOT necessary to do this for publication.

In Fig 6A., ∆che looks like it has anterior localization. Is this normal for ∆che?

It could be interesting to do scRNAseq on the activated TNFa-GFP+ liver cells to determine the cell type.

-------------

Reviewer #2: 

In this study, Wiles, Schlomann, et al investigate how motility affect gut colonization dynamics of Vibrio in a zebrafish model. Mutants defective for motility and chemotaxis were generated and analyzed in a variety of assays. In the zebrafish model, these mutants were unable to efficiently colonize and failed to outcompete another gut bacterium, Aeromonas. The motility and chemotaxis mutants were also found to primarily colonize the distal intestine, form aggregates, and were vulnerable to expulsion by peristaltic movement. The authors also use a very clever genetic system to turn motility on and off during gut colonization; these experiments confirm their initial observations and ensure that the observed phentoypes were not due to founder effects. In addition, the authors found that induction of host responses was dependent on Vibrio to maintain a specific spatial organization; the chemotaxis and motility mutants were failed to induce host responses.

The main conclusion, that Vibrio needs motility to maintain a distinct spatial pattern of gut colonization to avoid expulsion by peristalsis is novel and should be of interest to a wide audience; these findings may apply to many other flagellated gut bacteria, possibly also in mammals. The experiments are well-controlled and the writing is easy to follow. I have listed some minor suggestions for the authors' consideration.

One of the strengths of this study is the careful and clever genetics, comparing mutants with very specific defects in chemotaxis and motility (that leave the flagellum intact). The authors might want to consider analyzing a mutant that is defective in producing flagella altogether. The aggregation phenotype observed in the mot and che mutants might be due to non-productive expression of flagella and a completely aflagellate mutant might not display the phenotypes observed with the mot and che mutants.

Minor Comments, typos, etc:

Line 99-129: The last two paragraphs of the introduction are mostly a summary of the findings presented later in the paper; I think this section can be condensed significantly without loss of clarity to the reader.

Consistent with standard nomenclature in the field, gene name/symbols should be italicized when denoting mutations.

Generally, the presentation of the data is outstanding. The only thing I was confused about was what was measured in Fig. 1C (first thought was Vibrio), maybe labelling the y axis with "Aeromonas population" would help.

The authors should consider citing previous work from the Salmonella field. This previous work, although on a pathogenic organism, is in line (and strengthens) the findings in the current paper. The Hardt lab had shown that Salmonella mutants deficient for chemotaxis are delayed for inducing inflammation (PMID: 15213159). The Baumler lab reported that energy taxis towards nitrate is important for invasion of Peyer's patches (PMID: 27435462).

-------------

Reviewer #3: 

The manuscript by Wiles et al demonstrates the importance of bacterial motility in colonization and persistence in the zebrafish intestine using bacterial motility mutants, conditionally induced motility mutants and complementation of motility in the mutant strains. In addition, their hypothesis that motility aids in gut persistence is further supported by the use of a zebrafish mutant that has reduced intestinal transport that rescues the persistence of the bacterial motility mutants. The work described here very cleverly combines ingenious molecular techniques, time-lapse microscopy and the transparent zebrafish model to support their hypotheses. While the study is somewhat data dense with all the figures, supplementary material and 10 (!) time-lapse videos, one could argue that most, if not all, of the presented data is needed. Overall, this is a very well written research study. However, there are a few issues that should be clarified to support the reported results.

1. Fig 2C and 2F, S1 and S2 mov, page 8, lines 196-212: While it is clear in the movies that the Δmot bacterial strain is not motile, it is not so clear that there is a difference in the WT and Δche in the areas of high fluorescence. Fig 2C WT points to an area of "dense population of motile and planktonic cells" while Fig 2C Δche points to an area that looks very similar that is labeled "dense aggregate of non-motile cells". In the movies you only see motile or highly fluorescent regions. Therefore, how was the graph in Fig 2F determined? Materials and methods does not really make this clear how this was determined.

2. Page 14, lines 390-395: What is the delineation between "foregut region" and "intestinal tissues"? The authors state that 54% of the tnfa-positive cells in the foregut were macrophages, but 93% of tnfa-positive cells that were associated with intestinal tissues were macrophages. Isn't the foregut considered part of the intestinal tissue? Please clarify what is meant here.

3. Page 14, lines 395-398, Figure 6D: The authors state that in contrast to tnfa-positive macrophages in the foregut, the tnfa-positive cells found in the liver area were not macrophages or neutrophils. However, the "tnfa-positive macrophage" that the arrowhead is pointing to the in Fig 6D, looks just like 3 other cells that should be "tnfa-positive macrophage" outside the foregut region, presumably associated with the liver. Can you clarify this discrepancy?

4. Discussion: One issue that did not seem to be addressed anywhere in the paper, but probably should have been in the discussion was how the lack of chemotactic ability would cause a similar defect in persistence as the motility mutant. The authors show that the Δche mutant was still mobile in the intestine, so it is not just motility that is causing the defect. How would the chemotaxis of the WT strain in the gut allow it to persist? What is the WT strain sensing and/or how is chemotaxis keeping the WT from aggregating? The authors should discuss possibilities as to why this would be the case.

-------------

Reviewer #4: 

Wiles et al investigate the biogeography of a Vibrio symbiont in the zebrafish model and demonstrate that its spatial organization is dependent on motility and chemotaxis. In particular, Vibrio's motile behavior was essential for enabling persistence in the foregut, and prevent expulsion.

Overall, this is a very elegant study, with clever experimental designs and genetic tools both in the bacterium and in the host side to investigate mechanisms of symbiosis. In particular, the beautiful images and videos and the use of inducible genetic switches to manipulate motility are very thoughtful approaches, and the results are convincing. The manuscript is also very well written, and will be of interested to a broad audience. The authors need to be commended for this very nice work.

I have two comments for the authors consideration:

1) The movies in WT zebrafish showing expulsion of the delta-mot and of the delta-che mutants are impressive. Are movies available from the ret-/- mutant animals, showing that the detal-mot and delta-che mutant are rescued? Also, Fig S2B only shows the delta-mot mutant, but not the delta-che mutant, in ret-/- mice. What is the phenotype of the delta-che mutant in these animals?

2) Ln 66: "much remains unknown about how motility and behaviors promote intestinal colonization and provide bacteria a competitive advantage." In the context of host-pathogen interaction, work with Salmonella has shown the importance of motility and chemotaxis to promote intestinal colonization, to access nutrient sources in the mucus layer, and to outcompete the gut microbiota. I think it would be important to discuss the work of Stecher and Hardt, and Rivera-Chavez and Baumler, in the context of this new study.

---

## [Editor Report · Decision Letter 2]

24 Feb 2020

Dear Dr. Guillemin,

On behalf of my colleagues and the Academic Editor, Dr. Janelle S. Ayres, I am pleased to inform you that we will be delighted to publish your Research Article in PLOS Biology. 

Early Version

PRESS 

Kind regards,

Krystal Farmer

Development Editor, 

PLOS Biology

on behalf of

Hashi Wijayatilake,

Managing Editor

PLOS Biology